# Calibrating Video Watch-time Predictions with Credible Prototype Alignment

Chao Cui [* 1 2]  Shisong Tang [* 1 2]  Fan Li [1]  Jiechao Gao [3]  Hechang Chen [4]

## Abstract

Accurately predicting user watch-time is crucial for enhancing user stickiness and retention in video recommendation systems. Existing watch-time prediction approaches typically involve transformations of watch-time labels for prediction and subsequent reversal, ignoring both the natural distribution properties of label and the *instance representation confusion* that results in inaccurate predictions. wo-stage method combining prototype learning and optimal transport for watch-time regression prediction, suitable for any deep recommendation model. Specifically, we observe that the watch-ratio (the ratio of watch-time to video duration) within same duration bucket exhibits a multimodal distribution. To facilitate incorporation into models, we use a hierarchical vector quantized variational autoencoder (HVQ-VAE) to convert the continuous label distribution into a high-dimensional discrete distribution, serving as credible prototypes for calibrations. Based on this, ProWTP views the alignment between prototypes and instance representations as a Semi-relaxed Unbalanced Optimal Transport (SUOT) problem, where the marginal constraints of prototypes are relaxed. And the corresponding optimization problem is reformulated as a weighted Lasso problem for solution. Moreover, ProWTP introduces assignment and compactness losses to encourage instances to cluster closely around their respective prototypes, thereby enhancing the prototype-level distinguishability. Finally, we conducted extensive experiments, demonstrating our consistent superiority in real-world application.

## 1. Introduction

The rapid growth of online-video services (e.g. YouTube and Hulu) and video-sharing platforms (e.g. TikTok and Douyin) has driven the increasing demand for personalized and high-quality content (Zhou et al., 2018; Tang et al., 2023). In video recommendation systems, user watch-time has become a key metric for measuring user engagement (Covington et al., 2016; Tang et al., 2022; Li et al., 2024). Accurately predicting user watch-time not only helps improve user stickiness and retention but also optimizes content distribution and resource allocation, thereby driving the growth of Daily Active Users (DAUs) on the platform (Lin et al., 2023; Zhan et al., 2022; Wu et al., 2024).

Existing methods for Watch-time Prediction (WTP) usually focus on designing specific loss functions or transforming watch-time labels in particular ways to train the model, aiming to improve performance. Weighted Logistic Regression (WLR) (Covington et al., 2016) treats WTP task as a weighted binary classification problem, approximating the expected watch-time by assigning weights to positive samples. Duration-Deconfounded Quantile-based (D2Q) model (Zhan et al., 2022) divides videos into different groups based on duration and employs traditional regression within each group to estimate the transformed watch-time. Tree-based Progressive Regression (TPM) (Lin et al., 2023) decomposes WTP into a series of ordinal classifications, leveraging a tree structure to model conditional dependencies.

However, those methods struggle to consistently maintain high predictive accuracy across different models. They overlook the natural distribution properties of labels—we observed that the watch ratio (i.e., the ratio of watch-time to video duration) within the same video duration bucket exhibits a pronounced multimodal distribution, as shown in Fig.1(a), which has not yet been explicitly captured. Moreover, model trained with watch-time supervision suffers from *instance representation confusion*, as shown in Fig.1(b), making it challenging to accurately differentiate various patterns, consequently, limiting its predictive capability. To address the aforementioned issues, we propose a two-stage method called ProWTP, which combines prototype learning (Snell et al., 2017; Chang et al., 2022) and optimal transport (Villani et al., 2009; Peyré et al., 2019; Caffarelli & McCann, 2010; Chizat et al., 2018; Chapel

---
[*]Equal contribution  [1]Kuaishou Inc.  [2]Tsinghua University  [3]Stanford University  [4]Jilin University. Correspondence to: Hechang Chen <chenhc@jlu.edu.cn>.

*Proceedings of the 42nd International Conference on Machine Learning*, Vancouver, Canada. PMLR 267, 2025. Copyright 2025 by the author(s).

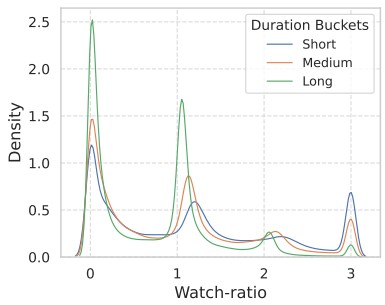

(a) Watch-ratio distribution.

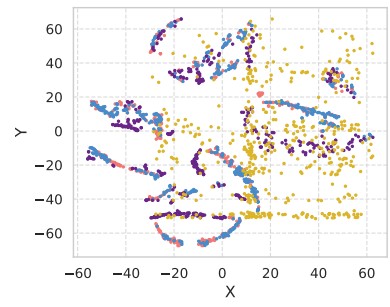

(b) Representation confusion.

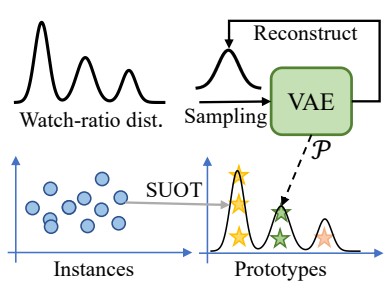

(c) Core idea of ProWTP.

*Figure 1.* (a) Illustrates the watch-ratio distribution of three different video durations, demonstrating the multimodal nature. (b) Depicts the instance representation confusion problem, where MLP serves as the deep recommendation model. (c) Shows the core idea of proposed ProWTP.

et al., 2021), making it applicable to any deep recommendation model. The core concept of ProWTP is illustrated in Fig.1(c), wherein instance distributions are aligned with credible label distributions to calibrate the instance representation space, thereby enhancing prediction accuracy. In the first stage, we employ a Hierarchical Vector Quantised Variational Auto-Encoder (HVQ-VAE) (Van Den Oord et al., 2017) to transform the one-dimensional continuous distribution of watch-ratio into a high-dimensional discrete distribution, generating credible prototypes that effectively capture the patterns of multimodal distributions of different duration buckets. Different from traditional prototype learning (Snell et al., 2017; Yang et al., 2023; Chang et al., 2022), ProWTP generates prototype vectors from label distributions, providing models with more precise and credible calibration references. Subsequently, we model the alignment between prototypes and instance representations as a Semi-relaxed Unbalanced Optimal Transport (SUOT) problem (Chapel et al., 2021), wherein the marginal constraints on the prototypes are relaxed. By reformulating the SUOT with an $l_2$ penalty term into a weighted Lasso regression problem, we utilize a regularization path algorithm to compute the OT plan (Chapel et al., 2021). Moreover, to further enhance the model's discriminative capability, we introduce the assignment and compactness losses that encourage instances to cluster around their respective prototypes. Our contributions are summarized as follows:

- We propose a method named ProWTP for the WTP task, which addresses the instance representation confusion problem in deep recommendation models by aligning label distributions with instance representation distributions through optimal transport, thereby enhancing model prediction performance.

- We investigate the multimodal distribution properties of watch-ratio across different video duration buckets for the first time and utilize the hierarchy VQ-VAE to

transform these into credible high-dimensional prototype vectors, providing a more precise reference for recommendation models calibration.

- We conducted extensive offline experiments on three industrial datasets and the experimental results consistently demonstrate the superiority of our approach.

## 2. Related Work

**Watch-time Prediction.** Watch-time prediction is a critical issue in industrial recommender systems, especially for platforms focusing on short videos and movies. Despite its significance, there are only a few papers that address this area (Lin et al., 2023; Covington et al., 2016; Zhan et al., 2022). A pioneering study (Covington et al., 2016) in YouTube's video recommendation sphere introduced the Weighted Logistic Regression (WLR) technique for forecasting watch durations. It has since been established as a leading method in related application areas. Nevertheless, this approach is not directly applicable to full-screen video recommendation systems and may encounter significant bias issues due to its weighting strategy. D2Q (Zhan et al., 2022) addresses duration bias by utilizing backdoor adjustment techniques and models watch time through direct quantile regression of viewing durations. Debiased and Denoised watch time Correction ($D^2Co$) (Zhao et al., 2023) and Counterfactual Watch Model (CWM) (Zhao et al., 2024) leverage causal inference frameworks, while Debiased Video Recommendation (DVR) (Zheng et al., 2022) employs adversarial learning to mitigate duration bias. SWAT (Yang et al., 2024) leverages a user-centric statistical framework with behavior-driven assumptions and bucketization techniques to model watch time. However, those method fail to consider the ordinal relationships and dependencies between different quantiles. Additionally, since both approaches estimate watch time using point estimations, they overlook the uncertainty inherent in the predictions. Then, TPM (Lin et al.,

2023) introduced the ordinal ranks of watch time and decomposed the problem into a series of conditional dependent classification tasks organized into a tree structure.

**Optimal Transport.** Optimal Transport (OT) (Villani et al., 2009; Peyré et al., 2019) is a mathematical tool used to transfer or match distributions. OT has been employed in a wide range of tasks including generative adversarial training (Arjovsky et al., 2017), clustering (Ho et al., 2017), domain adaptation (Courty et al., 2017), and others. Partial Optimal Transport (POT) (Caffarelli & McCann, 2010; Figalli, 2010) is an extension of the classical OT problem, where only a partial amount of mass is transported instead of transporting all the mass between two distributions. To alleviate the computational load of OT, the Sinkhorn algorithm (Cuturi, 2013) was introduced as an efficient method for solving Sinkhorn OT, and it was subsequently extended to POT (Benamou et al., 2015). Previously, many methods (Flamary et al., 2016; Damodaran et al., 2018) applied OT to domain adaptation, aligning the distributions of source and target domains in either input or feature spaces. They utilized mini-batch OT to mitigate computational overhead but faced sampling bias since mini-batch data only partially reflect the original data distribution. To tackle these challenges, more robust OT models, such as unbalanced and partial mini-batch OT, have been developed to enhance performance (Nguyen et al., 2022). Building on this, joint partial optimal transport was designed to transport only a portion of the mass, mitigating negative transfer, and the method was later applied to open-set domain adaptation (Xu et al., 2020). Additionally, aligning source prototypes with target features has been proposed as a solution to the problem of universal domain adaptation (Yang et al., 2023).

**Deep clustering with VAE.** Variational Autoencoders (VAEs) (Kingma, 2013) have emerged as a pivotal approach in the domain of deep clustering for unsupervised learning tasks, effectively overcoming the limitations of traditional clustering methodologies that often struggle with complex and high-dimensional data. By optimizing the evidence lower bound (ELBO), VAEs facilitate the learning of data embeddings while integrating prior knowledge, such as Gaussian Mixture Models (GMMs) (McLachlan et al., 2019), for modeling latent variables. Notable contributions in this field include the Variational Deep Embedding (VaDE) (Jiang et al., 2016) framework, which combines VAEs with GMMs, employing mixtures of Gaussian priors to enhance clustering performance. GMVAE (Dilokthanakul et al., 2016) addresses the problem of over-regularization in VAE by employing the minimum information constraint. LTVAE (Li et al., 2018) improves clustering by integrating a latent tree model into a VAE variant, introducing a tree-structured layer of discrete latent variables optimized via message passing. VAEIC (Prasad et al., 2020) jointly learns the prior and posterior parameters, thus avoiding pre-

training. The Vector Quantized Variational Autoencoder (VQ-VAE) (Van Den Oord et al., 2017) is an extension of the traditional VAE framework, which introduces a discrete latent space via a codebook of prototype vectors. In VQ-VAE, continuous latent vectors are quantized by mapping each to its closest prototype vector from the codebook, thus discretizing the latent representation. Although the quantization process is non-differentiable, techniques such as the Straight-Through Estimator (STE) (Yin et al., 2019) and Gumbel-Softmax (Jang et al., 2016) enable end-to-end training by allowing gradient-based optimization. The prototype vectors can serve as cluster centroids (Zheng & Vedaldi, 2023; Wu & Flierl, 2020), encapsulating essential information about distinct data clusters. In addition, the semantically rich prototypes learned by VQ-VAE can support various applications, such as conditional image generation (Esser et al., 2021; Ramesh et al., 2022), multi-modal language modeling (Li et al., 2023; Zhan et al., 2024) and recommender system (Liu et al., 2024; Rajput et al., 2024; Li & Sui, 2025).

## 3. Background

**Optimal Transport.** We consider two sets of data points, denoted as $\{x_i\}_{i=1}^n$ and $\{y_j\}_{j=1}^m$, where the empirical distributions are represented as $\boldsymbol{\mu} = \sum_{i=1}^n \mu_i \delta_{x_i}$ and $\boldsymbol{\nu} = \sum_{j=1}^m \nu_j \delta_{y_j}$, respectively. Here, $\sum_{i=1}^n \mu_i = 1$ and $\sum_{j=1}^m \nu_j = 1$, with $\delta_x$ indicating the Dirac delta function at location $x$. For simplicity in notation, we write $\boldsymbol{\mu} = (\mu_1, \mu_2, \ldots, \mu_n)^\top$ and $\boldsymbol{\nu} = (\nu_1, \nu_2, \ldots, \nu_m)^\top$, and define the cost matrix as $\mathbf{C} \in \mathbb{R}^{n \times m}$, where each element is $\mathbf{C}_{ij} = d(x_i, y_j)$. The Optimal Transport (OT), as defined by (Villani et al., 2009; Peyré et al., 2019), is a mathematical framework that transports a probability measure $\boldsymbol{\mu}$ into another measure $\boldsymbol{\nu}$ with a minimum cost $\mathbf{C}$. This can be formulated as the following linear programming problem:

$$\text{OT}(\boldsymbol{\mu}, \boldsymbol{\nu}) = \min_{\mathbf{T} \in \Pi(\boldsymbol{\mu}, \boldsymbol{\nu})} \langle \mathbf{T}, \mathbf{C} \rangle, \tag{1}$$

where $\langle \cdot, \cdot \rangle$ is the Frobenius dot product, $\mathbf{T} \in \mathbb{R}_{\geq 0}^{n \times m}$ is the transport plan. $\Pi(\boldsymbol{\mu}, \boldsymbol{\nu}) = \{\mathbf{T} \in \mathbb{R}_{\geq 0}^{n \times m} | \mathbf{T} \mathbf{1}_m = \boldsymbol{\mu}, \mathbf{T}^T \mathbf{1}_n = \boldsymbol{\nu}\}$ denotes the polytope of matrices $\mathbf{T}$.

**Unbalanced Optimal Transport.** However, the strict mass-conservation constraints on the transport plan $\mathbf{T}$ might cause dreadful degradation of performance in some applications. These constraints can be alleviated by incorporating the penalty of $\Pi(\boldsymbol{\mu}, \boldsymbol{\nu})$ into the objective function, which naturally leads to the formulation of the Unbalanced Optimal Transport (UOT) problem (Chizat et al., 2018; Chapel et al., 2021):

$$\text{UOT}^\lambda(\boldsymbol{\mu}, \boldsymbol{\nu}) = \min_{\mathbf{T} \geq 0} \langle \mathbf{T}, \mathbf{C} \rangle + \lambda_1 \Phi(\mathbf{T} \mathbf{1}_m, \boldsymbol{\mu}) + \lambda_2 \Phi(\mathbf{T}^T \mathbf{1}_n, \boldsymbol{\nu}), \tag{2}$$

where $\Phi(\cdot, \cdot)$ is a smooth divergence measure function, $\lambda_1$ and $\lambda_2$ are hyperparameters that represent the strengths of

penalization. We also have an alternative formulation, which relaxes one of the two constraints in (1). This is a Semi-relaxed Unbalanced Optimal Transport (SUOT) problem (Chapel et al., 2021), defined as the following:

$$\text{SUOT}^{\lambda}(\boldsymbol{\mu}, \boldsymbol{\nu}) = \min_{\mathbf{T} \geq 0, \mathbf{T}\mathbf{1}_m = \boldsymbol{\mu}} \langle \mathbf{T}, \mathbf{C} \rangle + \lambda \Phi(\mathbf{T}^T \mathbf{1}_n, \boldsymbol{\nu}) \quad (3)$$

**SUOT cast as regression.** Let $\mathbf{t} = \text{vec}(\mathbf{T})$ and $\mathbf{c} = \text{vec}(\mathbf{C})$. Next, we define matrices $\mathbf{H}_c$ and $\mathbf{H}_r$, such that $\mathbf{H}_c \mathbf{t}$ computes the column sums of the transport plan (i.e., $\mathbf{T}^{\top} \mathbf{1}_n$), and $\mathbf{H}_r \mathbf{t}$ computes the row sums (i.e., $\mathbf{T}\mathbf{1}_m$). The objective function for SUOT includes the transport cost $\langle \mathbf{C}, \mathbf{T} \rangle$ and the deviation penalty term $\lambda \Phi(\mathbf{T}^{\top} \mathbf{1}_n, \boldsymbol{\nu})$, where $\Phi$ is typically chosen as the squared Euclidean distance. Using vectorization and matrix notation, the objective function can be rewritten as $\mathbf{c}^{\top} \mathbf{t} + \lambda \|\mathbf{H}_c \mathbf{t} - \boldsymbol{\nu}\|_2^2$. Introducing the variable $\gamma = \frac{1}{\lambda}$, we reformulate the problem as:

$$\min_{\mathbf{t} \geq 0} \gamma \mathbf{c}^T \mathbf{t} + 0.5 * \|\mathbf{H}_c \mathbf{t} - \boldsymbol{\nu}\|_2^2, \quad s.t. \mathbf{H}_r \mathbf{t} = \boldsymbol{\mu}, \quad (4)$$

and as such be expressed as a non-negative penalized linear regression problem, where $\mathbf{H}_c \mathbf{t}$ is regressed onto the target distribution $\boldsymbol{\nu}$. By representing the SUOT problem in this form, we can leverage efficient optimization algorithms from regression analysis to solve it (Chapel et al., 2021).

# 4. Proposed Method: ProWTP

Let $\mathcal{U} = \{u_1, ..., u_{|\mathcal{U}|}\}$ and $\mathcal{V} = \{v_1, ..., v_{|\mathcal{V}|}\}$ denote the set of users and videos, respectively, where $|\mathcal{U}|$ is the number of users and $|\mathcal{V}|$ is the number of items. The user-item historical interactions are represented by $\mathcal{D} = \{(x_i, y_i) | x = (u, v), u \in \mathcal{U}, v \in \mathcal{V}\}_{i=1}^N$, where $N$ is the number of samples and $y \in \mathbb{R}^*$ denotes the watch-time. The target is to learn a deep recommendation model $f(X; \Theta_f)$ and a regressor $g(f(X; \Theta_f); \Theta_g)$ to predict the watch-time $y$ of user $u$ on video $v$, where $\Theta_f$ and $\Theta_g$ is the parameters of $f$ and $g$, respectively.

## 4.1. Overview

The proposed ProWTP is a two-stage method, as shown in Fig. 2. In the first stage, we employ a Hierarchical Vector Quantised Variational AutoEncoder (HVQ-VAE) $\mathbb{P}(\mathcal{P}|Y)$, which consists of three components: 1) Encoder $E(\cdot; \Theta_E) : \mathbf{Y} \to \mathbb{R}^d$ maps the one-dimensional continuous watch-ratio distribution $\mathbf{w} \in \mathbb{R}^L$ into a $d$-dimensional space, generating the initial representation $E(\mathbf{w}; \Theta_E) \in \mathbb{R}^d$; 2) Codebook $\mathcal{P} \in \mathbb{R}^{C \times K \times d}$: quantizes the high-dimensional feature into a discrete space, capturing the multimodal characteristics; 3) Decoder $D(\cdot; \Theta_D)$: $\mathbb{R}^d \to \hat{\mathbf{Y}}$ decodes the quantized prototype back into the continuous distribution $\mathbf{w}$, ensuring the reconstruction capability of the prototypes. In the second stage, the prototypes $\mathcal{P}$ and Semi-Relaxed Unbalanced Optimal Transport (SUOT) modules are integrated

to regularize the training of the recommendation model $f(\cdot; \Theta_f) : \mathbf{X} \to \mathbb{R}^d$ and calibrate the instance representation space, thereby producing accurate instance representations $\mathbf{h}$ for prediction by the regressor $g(\cdot; \Theta_g) : \mathbf{H} \to \mathbf{R}^*$.

## 4.2. Credible Prototypes Generation with HVQ-VAE

Currently, most prototype learning researches (Snell et al., 2017; Chang et al., 2022) typically rely on pre-trained models, where prototypes are generated by clustering the hidden representations for subsequent tasks. However, we argue that such prototypes often contain noise and potential errors, limiting their capacity in calibrating original models. Therefore, we propose to generate prototypes directly from the distribution of the prediction target $\mathbf{Y}$. As shown in Fig. 1(a), when we partition user historical behaviors into $\{1, 2, ..., D\}$ buckets based on video duration, we observe that the watch-ratio (i.e., the ratio of user's watch-time to video duration) within each bucket exhibits a distinct multimodal distribution. This indicates that user's behavior is statistically clustered and regular. However, these multimodal distributions are one-dimensional long sequences, making it challenging to directly extract high-dimensional discrete representations.

**Pre-processing.** To solve this problem, we first sample $L$ ($L >> D$) one-dimensional distributions $\mathbf{w} = (y_1, ..., y_n)$ from each multimodal distribution. Using this sampling strategy, we transform the original one-dimensional multimodal distributions into $D * L$ one-dimensional near-Gaussian distributions $\mathbf{w}$ of length $n$, thereby making the data more suitable for neural networks and effectively reducing the difficulty of training.

**Credible Prototypes Generation.** We observed in Fig. 1(a) that the peaks at the same positions across different duration buckets exhibit similar means but varying variances. Then, we hypothesize that $\mathbf{w}$ sampled from the same positions in these buckets can be grouped into equal means but varied variances clusters. Inspired by Vector Quantised Variational AutoEncode (VQ-VAE) (Van Den Oord et al., 2017), we propose a Hierarchical VQ-VAE approach that first identifies the closest cluster and then indexes the nearest vector within that cluster. Specifically, we take the one-dimensional distribution $\mathbf{w}$ into the encoder $E(\cdot; \Theta_E)$ to obtain latent representation $E(\mathbf{w})$. Subsequently, the HVQ-VAE maintains a codebook $\mathcal{P} \in \mathbb{R}^{C \times K \times d}$, where $C$ and $K$ are the number of cluster and the number of prototype, respectively, $\mathbf{p}_{ij} \in \mathbb{R}^d$ is a prototype vector. We can assume that prototype vectors within the same cluster share similar means but allow for different variances.

Next, we select the cluster $c$ by computing the distance between $E(w)$ and each cluster center $\tilde{\mathbf{p}}_i$, where $\tilde{\mathbf{p}}_i$ is obtained by target attention (Zhou et al., 2018; Vaswani, 2017)

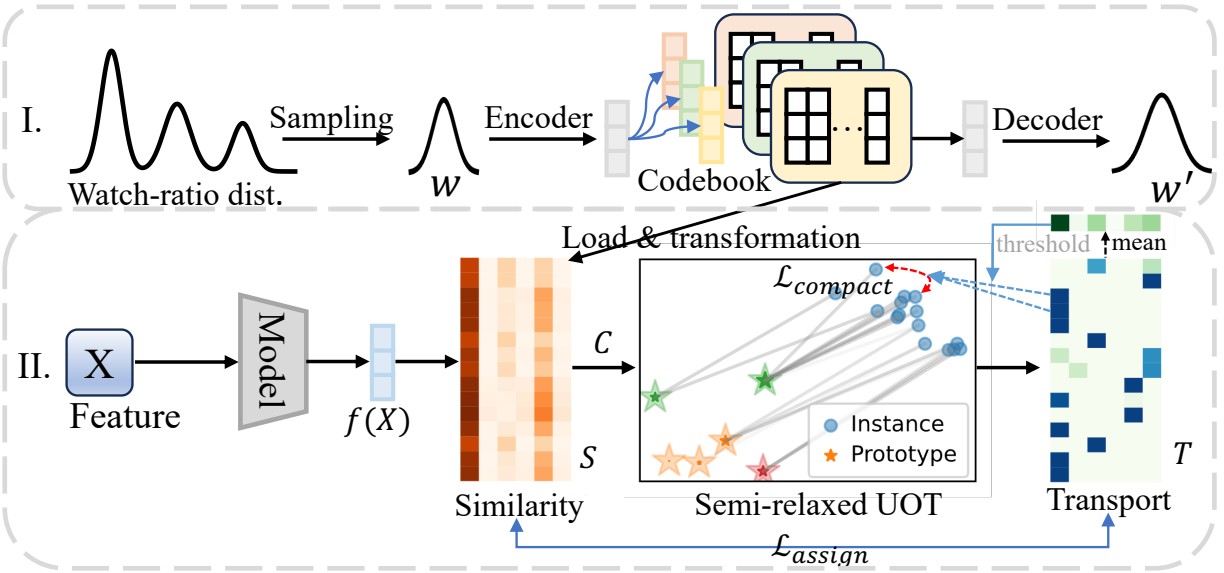

*Figure 2.* The framework of proposed ProWTP, which contains two phases: credible prototypes generation and distribution alignment. In the first stage, HVQ-VAE is used to encode the watch-ratio distribution into high-dimensional discrete representations, which serve as prototypes for calibration. In the second stage, semi-relaxed unbalanced optimal transport (SUOT) is employed to align the instance distribution with the prototypes, thereby calibrating the instance space.

between $E(\mathbf{w})$ and $\mathcal{P}$:

$$c = \operatorname{argmin}_i ||\tilde{\mathbf{p}}_i - E(\mathbf{w})||_2,$$
$$\text{where} \quad \tilde{\mathbf{p}}_i = \sum_j \operatorname{softmax}\left(\mathbf{p}_{ij} \cdot E(\mathbf{w})\right) \cdot \mathbf{p}_{ij}. \quad (5)$$

Within the selected cluster $c$, we find the prototype vector $\mathbf{p}_{c,k}$ that has the minimum distance to $E(\mathbf{w})$ and map it to the discrete vector $\mathbf{z}$:

$$\mathbf{z} = \mathbf{p}_{c,k}, \quad \text{where} \quad k = \arg\min_j ||\mathbf{p}_{c,j} - E(\mathbf{w})||_2 \quad (6)$$

Finally, we input $\mathbf{z}$ into the decoder $D(\cdot)$ to reconstruct $\mathbf{w}$. In HVQ-VAE, the presence of the argmin operation hampers gradient propagation. To address this issue, we employ the Straight-Through Estimator (STE) (Bengio et al., 2013; Van Den Oord et al., 2017) during training, with the loss function defined as follows:

$$\mathcal{L}_{\text{HVQ-VAE}} = ||\mathbf{w} - D(E(\mathbf{w}) + \operatorname{sg}[\mathbf{z} - E(\mathbf{w})])||_2^2$$
$$+ ||\operatorname{sg}[E(\mathbf{w})] - \mathbf{z}||_2^2 + \beta ||E(\mathbf{w}) - \operatorname{sg}[\mathbf{z}]||_2^2 \quad (7)$$

Here, sg[·] denotes the stop-gradient operation, which halts gradient flow during backpropagation, and $\beta$ is a hyperparameter that balances the reconstruction loss and the embedding update. Through this approach, we transform the one-dimensional continuous $w$ distribution into discrete high-dimensional prototype vectors, thereby providing credible calibration for subsequent models.

## 4.3. distribution alignment

As illustrated in Fig. 1(b), we posit that the inaccuracies of recommendation models within the WTP task stem from *instance representation confusion*. This confusion hampers the model's ability to effectively differentiate between various user behavior patterns, thereby adversely affecting predictive performance. To address this issue, it is imperative to utilize the generated credible prototypes $\mathcal{P}$ to calibrate the instance representation $f(x)$, thereby reducing representation confusion and enhancing the model's predictive accuracy.

**Transport Matrix Calculation.** First, we conceptualize the instance representations $f(x)$ and the prototypes $\mathcal{P}$ as two probability distributions, with the objective of mapping the instance representation distribution $\boldsymbol{\alpha} = \frac{1}{n_b}\mathbf{1}_{n_b}$ to the prototype representation distribution $\boldsymbol{\beta} = \frac{1}{CK}\mathbf{1}_{CK}$ through optimal transport (Villani et al., 2009; Peyré et al., 2019; Chapel et al., 2021). Specifically, we construct the instance representation set $\mathbf{H} = \{\mathbf{h}_1, \mathbf{h}_2, \ldots, \mathbf{h}_{n_b}\} \subseteq \mathbb{R}^d$, where each instance representation $\mathbf{h}_i$ is obtained by $L_2$ normalization of the model output $f(x_i)$, i.e. $\mathbf{h}_i = f(x_i)/||f(x_i)||_2$, and $n_b$ is the mini-batch size. The prototype set $\mathbf{P} = \{\mathbf{p}_1, \mathbf{p}_2, \ldots, \mathbf{p}_{CK}\} \subseteq \mathbb{R}^d$ is derived from the original prototype set $\mathcal{P}$ through a learnable linear transformation $\mathbf{W}_p$. To quantify the discrepancy between instances and prototypes, we define the cost matrix $\mathbf{C} \in \mathbb{R}^{n_b \times CK}$, where each element $c_{i,k}$ represents the cosine distance between instance $\mathbf{h}_i$ and prototype $\mathbf{p}_k$, i.e. $\mathbf{C} = 1 - \mathbf{H}^T * \mathbf{P}$.

To achieve distribution alignment, we adopt the optimal transport method. However, traditional optimal transport requires all the mass from $\boldsymbol{\beta}$ is transported to $\boldsymbol{\alpha}$, meaning that each prototype must be fully mapped to the instances. This strict marginal constraint is not applicable in our scenario, especially in a mini-batch setting, where it is unreasonable to allocate samples for every prototype, as certain prototypes may not correspond to any instances in the current batch. Therefore, we model the alignment between instances and prototypes as a Semi-relaxed Unbalanced Optimal Transport (SUOT) problem (Chapel et al., 2021):

$$\mathbf{T}^* = \text{SUOT}^\lambda(\boldsymbol{\alpha}, \boldsymbol{\beta}) = \min_{\mathbf{T} \geq 0, \mathbf{T}\mathbf{l}_{CK}=\boldsymbol{\alpha}} \langle \mathbf{T}, \mathbf{C} \rangle + \lambda ||\mathbf{T}^T \mathbf{l}_{n_b} - \boldsymbol{\beta}||_2^2,$$
(8)

where $\lambda$ controls the strengths of penalization. By introducing an $l2$ penalty term into the objective, we allow the marginal constraints on the prototype side to be relaxed, transforming the hard constraints $\mathbf{T}^T \mathbf{l}_{n_b} = \boldsymbol{\beta}$ into soft one. To optimize this problem, (Chapel et al., 2021) reformulated it as a weighted Lasso regression and solved it with a regularization path algorithm.

**Training objectives.** To calibrate the sample space, we aim for the instance representations to cluster tightly around their corresponding prototypes, necessitating a reduction in the distance between each sample and its assigned prototype. Each row of the transportation matrix $\mathbf{T}$ represents the allocation relationship of sample $x_i$ to the prototypes, with the row sums equal to $\frac{1}{C \times K}$. After multiplying by the constant $C \times K$, each row of $\mathbf{T}$ can be viewed as a soft pseudo-label summing to 1. Therefore, we can define the calibration loss through the cross-entropy loss:

$$\mathcal{L}_{assign} = -\frac{1}{n_b} \sum_{i=1}^{n_b} \sum_{k=1}^{C*K} t_{i,k} \log \frac{\exp(\mathbf{h}_i^T * \mathbf{p}_k / \tau)}{\sum_{j=1}^{C*K} \exp(\mathbf{h}_i^T * \mathbf{p}_j / \tau)}$$
(9)

where $\tau$ is the temperature parameter that controls the smoothness of the softmax function. By minimizing $\mathcal{L}_{assign}$, we can decrease the distance between samples and their corresponding prototypes, thereby better calibrating instance representations within the prototype space, reducing representation confusion, and enhancing the model's predictive performance.

To further shape the instance space, we hope for instances assigned to the same prototype to be closer together in the representation space, thereby forming tighter clusters. This necessitates promoting similarity among samples under the same prototype. To achieve this, we first define the set of instances associated with each prototype $k$:

$$\mathcal{S}_k^+ = \{i | t_{i,k} > \frac{1}{n_b} \sum_j t_{j,k}\},$$
(10)

which includes those samples under prototype $\mathbf{p}_k$ whose transport value $t_{i,k}$ exceed the average level, indicating

that these samples should be close to each other in the instance representation space. Inspired by contrastive learning (Khosla et al., 2020), we designed a compact loss to encourage samples under the same prototype to cluster more closely in the representation space:

$$\mathcal{L}_{\text{compact}} = -\frac{1}{CK} \sum_{k=1}^{CK} \sum_{i,j=1}^{n_b} \mathbb{I}(i, j \in \mathcal{S}_k^+) \cdot \mathbb{I}(i \neq j)$$
$$\cdot \log \frac{\exp(\mathbf{h}_i^T \mathbf{h}_j / \tau)}{\sum_{i',j'=1}^{n_b} \exp(\mathbf{h}_{i'}^T \mathbf{h}_{j'} / \tau)},$$
(11)

where $\mathbb{I}(\cdot)$ is the indicator function, and $\tau$ controls the smoothness. By minimizing the compact loss, we not only help reduce instance representation confusion but also enhance the model's ability to capture fine-grained features, ultimately improving prediction performance. Additionally, to address computational efficiency issues arising from multiple loops, we randomly sample $20\%$ of the instances from the mini-batch for the calculations. Finally, We incorporate the labels $y_i$ to define the MSE loss:

$$\mathcal{L}_{task} = \frac{1}{N} \sum_{i=1}^{N} (g(\sum_{k=1}^{CK} t_{i,k} \mathbf{p}_k) - y_i)^2,$$
(12)

Compared to original prediction, ProWTP reshapes instance representations $f(x_i)$ in the credible prototype space $\mathbf{P}$ by utilizing the transport matrix $\mathbf{T}$ to weight and combine prototype vectors, subsequently feeding these representations into the regressor $g(\cdot; \Theta_g)$ for prediction. This approach effectively captures the inherent structure of the instance representation space, enhancing the model's robustness and leading to more accurate predictions.

## 5. Experiment

### 5.1. Setup

**Dataset.** We adopt two public datasets Wechat (collected from Wechat App) and Kuairand (Gao et al., 2022) (from Kuaishou App), one private dataset Short-video (from our App) for offline experiments. We split each dataset into training, validation and test set by the ratio of 6:2:2.

**Baselines.** We evaluate the performance of proposed ProWTP in comparison with the following baselines that represent the popular method in WTP tasks: Traditional Regression, Weighted Logistic Regression (WLR) (Covington et al., 2016), Ordinal Regression (OR) (Crammer & Singer, 2001), Duration-Deconfounded Quantile (D2Q) (Zhan et al., 2022), Tree-based Progressive Model (TPM) (Lin et al., 2023). Sine all meth ods are model-agnostic, Debiased Video Recommendation (DVR) (Zheng et al., 2022), and Counterfactual Watch Model (CWM) (Zhao et al., 2024), we implement them on the MLP (Taud & Mas, 2018).

*Table 1.* Overall performance of different methods. Boldface means the best-performed methods. Higher XAUC and XGAUC indicate better performance, while lower MAE and RMSE are better.

| Model | Metrics | TR | WLR | OR | D2Q | TPM | DVR | CWM | ProWTP |
|---|---|---|---|---|---|---|---|---|---|
| WeChat | RMSE | 30.39 | 30.24 | 28.96 | 29.12 | 28.85 | 28.91 | 28.78 | **28.47** |
| | MAE | 20.53 | 20.16 | 20.05 | 20.12 | 19.97 | 20.05 | 19.90 | **19.84** |
| | XAUC | 0.5985 | 0.6047 | 0.6078 | 0.6094 | 0.6107 | 0.6109 | 0.6115 | **0.6183** |
| | XGAUC | 0.5409 | 0.5538 | 0.5575 | 0.5616 | 0.5645 | 0.5628 | 0.5654 | **0.5730** |
| KuaiRand-Pure | RMSE | 42.41 | 42.17 | 41.44 | 41.65 | 40.82 | 40.97 | 40.75 | **40.45** |
| | MAE | 28.09 | 27.98 | 27.69 | 27.82 | 24.58 | 26.08 | 24.54 | **24.43** |
| | XAUC | 0.7176 | 0.7081 | 0.7145 | 0.7189 | 0.7203 | 0.7201 | 0.7209 | **0.7290** |
| | XGAUC | 0.6907 | 0.6885 | 0.6945 | 0.6990 | 0.7024 | 0.6998 | 0.7026 | **0.7048** |
| Short-video | RMSE | 30.59 | 30.22 | 29.18 | 29.35 | 29.02 | 29.18 | 29.00 | **28.67** |
| | MAE | 11.46 | 11.29 | 11.07 | 11.15 | 10.82 | 11.08 | 10.76 | **10.69** |
| | XAUC | 0.5744 | 0.5788 | 0.5705 | 0.5814 | 0.5831 | 0.5822 | 0.5848 | **0.5929** |
| | XGAUC | 0.5537 | 0.5603 | 0.5609 | 0.5622 | 0.5667 | 0.5643 | 0.5681 | **0.5731** |

**Evaluation.** To evaluate the performance of each model, we use four widely adopted metrics (Zhan et al., 2022) : MAE, RMSE, XAUC, and XGAUC.

**Training details.** We set the embedding dimension of all features to 64. For TR and OR, models are implemented on MLP with three hidden layers and ReLU (Glorot et al., 2011) as the activation function. For other baseline methods, we adopt the experimental design and parameter settings described in the original papers. We optimize all models using Adam optimizer (Kingma & Ba, 2014) with the batch size of 512 on both two datasets. To avoid overfitting, We set the dropout rate (Srivastava et al., 2014) to 0.2 and employ an early stopping mechanism (Prechelt, 2002) with a patience of 10 epochs. Among them, the learning rate is searched in $\{1e\text{-}3, 1e\text{-}4, 1e\text{-}5\}$, and $\beta$ is tuned from 0.0 to 0.2 with increments of 0.05. $K$ is searched in $\{4, 8, 12, 16, 20, 24\}$.

### 5.2. Results

**Comparison with baselines.** We compare ProWTP with several baseline methods on two real-world industrial-grade datasets, and the results are shown in Tab. 1. ProWTP achieves the best performance across all evaluation metrics. In contrast, the TR method performs the worst on both datasets, likely because it directly regresses on watch-time without leveraging the distributional characteristics of the data. WLR and OR show some improvement over TR, but the gains are limited. The D2Q, by addressing duration bias, improves prediction accuracy, and TPM further enhances performance through its tree-structured modeling of dependencies and uncertainties. ProWTP outperforms all baselines in four metrics, particularly with significant improvements in RMSE and XAUC. This demonstrates that ProWTP effectively alleviates instance representation confusion by aligning the credible prototype distribution with the instance distribution, improving model's accuracy.

**Impact of different modules in ProWTP.** We further conduct ablation studies to demonstrate the effectiveness of the key components of ProWTP and the results are shown in Tab. 2. Specifically, we compare ProWTP to its five variants: (1) w/o HVQ-VAE, means that prototypes are no longer generated from label distributions but are randomly initialized as parameters within the neural network. (2) w/o $\mathcal{L}_{assign}$ means the assign loss is removed. (3) w/o $\mathcal{L}_{compact}$ means the compact loss is further removed. (4) w/o SUOT indicates that SUOT is no longer used for distribution alignment, and instead, the the linear combination of prototypes is directly computed for prediction. (5) w/o ProWTP means the approach degenerates into traditional regression (TR). The results indicate that removing any single module leads to a performance decline, demonstrating that each component of ProWTP is crucial for improving model performance. Removing HVQ-VAE results in a significant drop in performance, highlighting that transforming label distributions into credible prototypes effectively enhances the model's performance. The impact of removing SUOT is also particularly notable, indicating that SUOT helps better align the distributions of instances and prototypes, thereby improving predictive capabilities. Moreover, the two loss functions effectively constrain the learning of the instance space, ensuring instances are tightly clustered around the corresponding prototype, which enhances the model's discriminative ability.

**Different prototype generation methods.** To further validate the effectiveness of using HVQ-VAE for generating credible prototypes, we compare three different generation strategies: VQ-VAE, Kmeans, and Random. As shown in Tab. 3, the performance of VQ-VAE saw a slight decrease, indicating that the hierarchically generated prototypes from HVQ-VAE exhibit a better clustering structure, making it easier for instance representations to align with them. Kmeans generates prototypes by clustering the in-

*Table 2.* Ablation results of different modules in ProWTP.

| Model | Wechat | | | | KuaiRand-Pure | | | |
|---|---|---|---|---|---|---|---|---|
| | RMSE | MAE | XAUC | XGAUC | RMSE | MAE | XAUC | XGAUC |
| ProWTP | **28.47** | **19.84** | **0.6183** | **0.5730** | **40.45** | **24.43** | **0.7290** | **0.7048** |
| ProWTP w/o HVQ-VAE | 29.12 | 20.23 | 0.6128 | 0.5690 | 41.08 | 24.87 | 0.7233 | 0.7010 |
| ProWTP w/o $\mathcal{L}_{assign}$ | 29.45 | 20.65 | 0.6112 | 0.5678 | 41.35 | 25.01 | 0.7205 | 0.6998 |
| ProWTP w/o $\mathcal{L}_{compact}$ | 29.38 | 20.51 | 0.6130 | 0.5684 | 41.12 | 24.92 | 0.7221 | 0.7004 |
| ProWTP w/o SUOT | 29.90 | 20.89 | 0.6108 | 0.5665 | 42.00 | 25.50 | 0.7185 | 0.6980 |
| w/o ProWTP | 30.39 | 20.51 | 0.5979 | 0.5406 | 42.41 | 28.09 | 0.7174 | 0.6905 |

*Table 3.* Ablation study on different prototype generation methods.

| Prototypes generation | Wechat | | | | KuaiRand-Pure | | | |
|---|---|---|---|---|---|---|---|---|
| | RMSE | MAE | XAUC | XGAUC | RMSE | MAE | XAUC | XGAUC |
| HVQ-VAE | **28.47** | **19.84** | **0.6183** | **0.5730** | **40.45** | **24.43** | **0.7290** | **0.7048** |
| VQ-VAE | 28.82 | 20.04 | 0.6164 | 0.5713 | 40.72 | 24.52 | 0.7259 | 0.7024 |
| Kmeans | 29.07 | 20.28 | 0.6132 | 0.5683 | 41.22 | 24.98 | 0.7236 | 0.7018 |
| Random | 29.12 | 20.23 | 0.6128 | 0.5690 | 41.08 | 24.87 | 0.7233 | 0.7010 |

*Table 4.* Ablation study on different distribution alignment methods.

| Distribution alignment | Wechat | | | | KuaiRand-Pure | | | |
|---|---|---|---|---|---|---|---|---|
| | RMSE | MAE | XAUC | XGAUC | RMSE | MAE | XAUC | XGAUC |
| SUOT | **28.47** | **19.84** | **0.6183** | **0.5730** | **40.45** | **24.43** | **0.7290** | **0.7048** |
| OT | 28.82 | 20.15 | 0.6164 | 0.5705 | 40.85 | 24.65 | 0.7252 | 0.7023 |
| UOT | 29.46 | 20.58 | 0.6137 | 0.5688 | 41.20 | 24.93 | 0.7225 | 0.7000 |
| w/o alignment | 29.90 | 20.89 | 0.6108 | 0.5665 | 42.00 | 25.50 | 0.7185 | 0.6980 |

stance representations directly, but its performance drops due to being more susceptible to noise and potential errors. The Random method performs the worst, as it fails to provide a credible reference for calibrations, thereby affecting the model's predictive performance.

**Different distribution alignment methods.** We also compare different alignment strategies, as shown in Tab 4. The SUOT approach, which relaxes the marginal constraints on the prototype side, yielded the best performance. In contrast, OT requires strict transportation of the entire mass, but since not all prototypes in a mini-batch can be assigned to instances, it limits performance. UOT also saw a performance decline due to some instances not being assigned. SUOT's flexible allocation mechanism more effectively enhances model performance.

**Impact of the number of prototypes $K$.** In Fig 3, we illustrate the impact of varying the number of prototypes $K$ on model performance. As the number of prototypes increases, performance improves accordingly. However, defining too many prototypes results in slight performance fluctuations, likely due to the introduction of noise.

## 6. Conclusion

In this paper, we propose a two-stage method, ProWTP, for watch-time prediction (WTP) tasks, applicable to any deep recommendation model. This method aligns label distributions with instance representation distributions through prototype learning and optimal transport to calibrate the instance space, thereby improving the accuracy. Specifically, we employ HVQ-VAE to transform continuous watch-ratio labels into high-dimensional discrete distributions, which serve as credible prototypes. Then, the alignment between prototypes and instance representations is modeled as a SUOT problem, where the marginal constraints are relaxed and the problem is reformulated as a weighted Lasso regression for solution. Additionally, we introduce assign loss and compact loss to encourage instances to cluster tightly around their respective prototypes. Finally, extensive experiments demonstrate the significant advantages of ProWTP in practical applications.

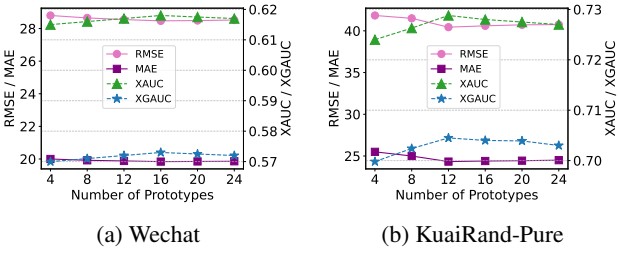

(a) Wechat                    (b) KuaiRand-Pure

*Figure 3.* Impact of the number of prototypes $K$ on two datasets.

## Impact Statement

This work introduces ProWTP, a calibrated watch-time prediction framework that aligns model representations with credible label-derived prototypes using semi-relaxed unbalanced optimal transport. It brings clear practical benefits to industrial video recommendation systems by improving prediction accuracy, reducing representation confusion, and better modeling the multimodal nature of user behavior. These improvements can enhance user experience, content exposure fairness, and platform efficiency. However, like all engagement optimization techniques, ProWTP may raise concerns about potential over-personalization or reinforcement of addictive behaviors. Nonetheless, since the method focuses on improving calibration fidelity rather than increasing engagement directly, we believe its responsible deployment can lead to more transparent and equitable recommendation systems.

## Acknowledgments

This work is supported in part by the National Natural Science Foundation of China (No. 62476110, No. U2341229); the NSF under grants III-2106758 and POSE-2346158; the National Key R&D Program of China (No. 2021ZD0112500); the Key R&D Project of Jilin Province (No. 20240304200SF); and the International Cooperation Project of Jilin Province (No. 20220402009GH).

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

# A. Appendix

## A.1. explanations on instance representation confusion.

### A.1.1. MATHEMATICAL EXPLANATION

**Proposition A.1.** *In WTP task, let the instance representation of a sample $(x, y)$ be $f(x)$, with its ideal center being $\mu_y = \mathbb{E}[f(x) \mid y]$, where $y$ is the ground-truth. The degree of instance representation confusion is defined as the distance between the instance representation and the ideal center, $d(f(x), \mu_y) = \|f(x) - \mu_y\|$. Then, the model's prediction error $\Delta_x = |y - \hat{y}|$ is predominantly correlated with the degree of instance representation confusion $d(f(x), \mu_y)$.*

*Proof*:

In a regression model for WTP task, suppose the predicted value is given by $\hat{y} = \mathrm{ReLU}(W f(x) + b)$, and the true value $y$ is a function represented by a ideal center $\mu_y = \mathbb{E}[f(x) \mid y]$ and a noise term $\epsilon$:

$$y = \mathrm{ReLU}(W \mu_y + b) + \epsilon, \tag{13}$$

where $W \in \mathbb{R}^{1 \times d}$ and $b \in \mathbb{R}$ are the model parameters, and $f(x) \in \mathbb{R}^d$ is the instance representation of the input $x$. The noise $\epsilon$ is independent and identically distributed Gaussian noise with zero mean, unrelated to the instance representation, i.e., $\epsilon \sim \mathcal{N}(0, \sigma^2)$.

Starting with the model and true value definitions, the error can be rewritten as:

$$\Delta_x = |y - \hat{y}| = |\mathrm{ReLU}(W \mu_y + b) + \epsilon - \mathrm{ReLU}(W f(x) + b)|. \tag{14}$$

We assume that the ideal center $\mu_y$ lies within the activation region, meaning that $W \mu_y + b \geq 0$. This assumption holds because, in WTP task, the ground-truth $y \geq 0$. Thus, we only need to consider two cases based on the value of $W f(x) + b$ for each sample:

**(1) Case 1**: $W f(x) + b \geq 0$

In the linear activation region of ReLU, the output simplifies to:

$$\Delta_x = |(W \mu_y + b + \epsilon) - (W f(x) + b)|. \tag{15}$$

Further simplifying:

$$\Delta_x = |W(\mu_y - f(x)) + \epsilon|. \tag{16}$$

The squared error is:

$$\Delta_x^2 = (W(\mu_y - f(x)))^2 + 2\epsilon W(\mu_y - f(x)) + \epsilon^2. \tag{17}$$

Taking the expectation, assuming $\epsilon$ is independent of $f(x)$ and $\mathbb{E}[\epsilon] = 0$:

$$\mathbb{E}[\Delta_x^2] = (W(\mu_y - f(x)))^2 + \mathbb{E}[\epsilon^2]. \tag{18}$$

Since $\mathbb{E}[\epsilon^2] = \sigma^2$, we have:

$$\mathbb{E}[\Delta_x^2] = (W(\mu_y - f(x)))^2 + \sigma^2. \tag{19}$$

Thus, the expectation of the squared error is dominated by $(W(\mu_y - f(x)))^2$, and we get:

$$(W(\mu_y - f(x)))^2 = \|W\|^2 \cdot \|f(x) - \mu_y\|^2. \tag{20}$$

Therefore:

$$\mathbb{E}[\Delta_x^2] \propto \|f(x) - \mu_y\|^2. \tag{21}$$

**(2) Case 2**: $W f(x) + b < 0$

In the non-activation region of ReLU, if $Wf(x) + b \leq 0$, then:

$$\hat{y} = 0. \tag{22}$$

In this case, the error is:

$$\Delta_x = |y|. \tag{23}$$

Combining both cases, the expected squared error is:

$$\begin{aligned}
\mathbb{E}[\Delta_x^2] = P(Wf(x) + b \geq 0) \cdot \left( \|W\|^2 \cdot \|f(x) - \mu_y\|^2 + \sigma^2 \right) \\
+ P(Wf(x) + b < 0) \cdot y^2.
\end{aligned} \tag{24}$$

When most instances satisfy $Wf(x) + b > 0$ (i.e., the ReLU activation region dominates), the expected error is primarily determined by $\|f(x) - \mu_y\|$. We conducted experiments and found that instances located in the non-activation region of ReLU account for approximately 1% to 2% of the total training data.

### A.1.2. DIFFERENT MODEL ANALYSIS

In this paper, we identify **instance representation confusion** as the main reason for the inability of existing methods to achieve accurate predictions. In Appendix A.1.1, we provide a mathematical explanation of the phenomenon. In this section, we conduct a visualization study on the relationship between instance representations $f(x)$ and prediction errors $\Delta$ across different values.

To simplify the analysis, we focus on three sample groups with true values $y \in [0, 0.1)$, $y \in [1.0, 1.1)$, and $y \in [2.0, 2.1)$. For each sample $x$, the prediction error of the model $f(\cdot)$ is denoted as $\Delta$. We define the ideal center $u_y$ as the average instance representation $f(x)$ of samples with $\Delta < 0.01$. The degree of instance representation confusion is measured by the $L_2$ distance $\|f(x) - u_y\|$.

The analysis results for each model include five figures: (a) The correlation between prediction error and the degree of confusion. (b) A t-SNE visualization of instance representations $f(x)$ for all three sample groups with ($\Delta < 0.3$). (c)(d)(e) The visualization of instance representations $f(x)$ and ideal centers $u_y$ for high-error samples ($\Delta > 0.3$) in $y \in [0, 0.1)$, $y \in [1.0, 1.1)$ and $y \in [2.0, 2.1)$ respectively.

From Figure (a), it can be observed that both TR and ProWTP align with the conclusion of Appendix A.1.1, where the prediction error $\Delta$ is positively correlated with the degree of confusion. From the distribution of black scatter points, TR exhibits a significantly higher level of confusion, while ProWTP effectively mitigates this confusion by reducing the distance between instances and reliable prototypes.

From Figure (b), even when the prediction error $\Delta < 0.3$, the instance representations of TR struggle to form well-defined clusters, with instances of different types mixed together. In contrast, ProWTP achieves clear clustering among instances with small errors, and instances of different types are distinctly separated.

In Figures (c), (d), and (e), for points with larger errors, darker colors indicate higher $\Delta$ values and greater distances from the ideal center. This further supports the conclusion in Appendix A.1.1. Additionally, compared to TR, ProWTP shows significantly fewer points with large errors (i.e., fewer dark-colored points), effectively reducing instance representation confusion. This demonstrates that the root cause of reducing prediction errors lies in learning better instance representations.

### A.2. More details of Pre-processing and Prototype generations.

Our goal is to transform the ground-truth $Y$ of the entire dataset into high-dimensional vectors, referred to as prototypes, for downstream tasks. The generation process is divided into four steps, as shown in Fig. 6 :

1. **Partitioning $Y$.** The ground-truth $Y$ is an $N \times 1$ vector, where $N$ is the size of the dataset. Directly generating prototypes from this vector is challenging. We observe that watch-ratio in different duration buckets exhibit distinct multi-modal distributions. Thus, $Y$ is first divided into $D$ unequal-length multi-modal distributions based on video durations.

2. **Fitting Gaussian Mixture Models (GMMs).** Even after partitioning, the watch-ratio distributions remain as long one-dimensional continuous arrays, making direct modeling still difficult. To address this, we fit $D$ GMMs to these distributions, where the number of components $C$ corresponds to the number of peaks in the distribution.

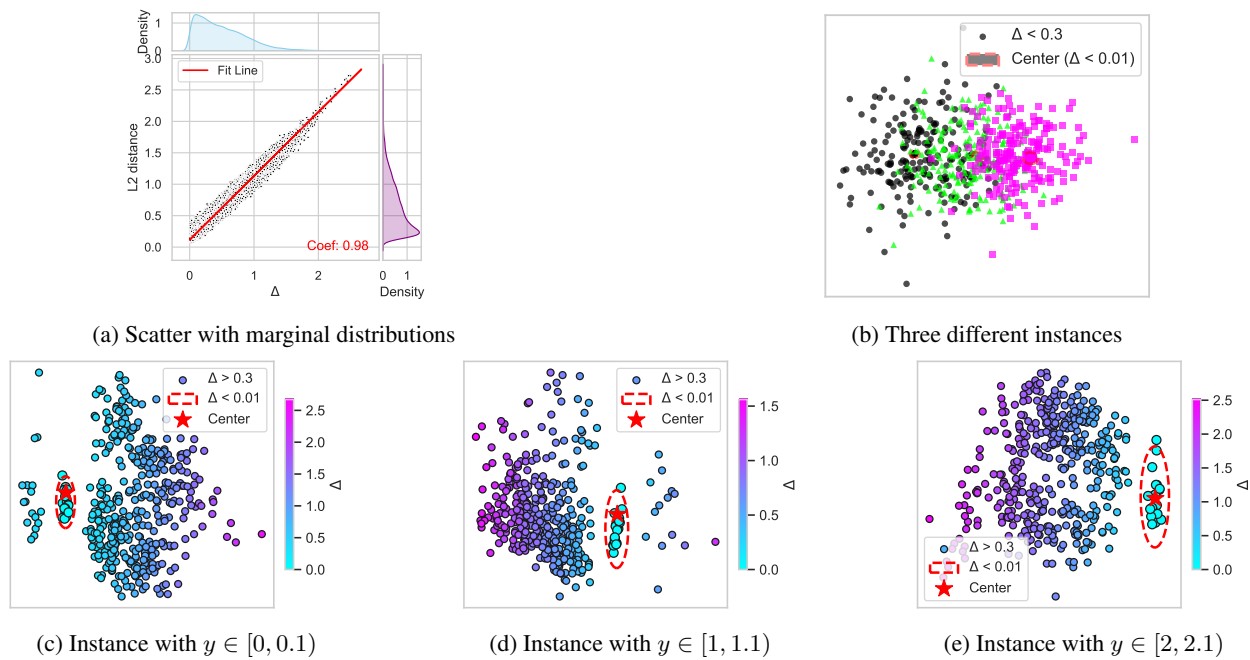

(a) Scatter with marginal distributions

(b) Three different instances

(c) Instance with $y \in [0, 0.1)$

(d) Instance with $y \in [1, 1.1)$

(e) Instance with $y \in [2, 2.1)$

*Figure 4.* Traditional Regression (TR) on Wechat.

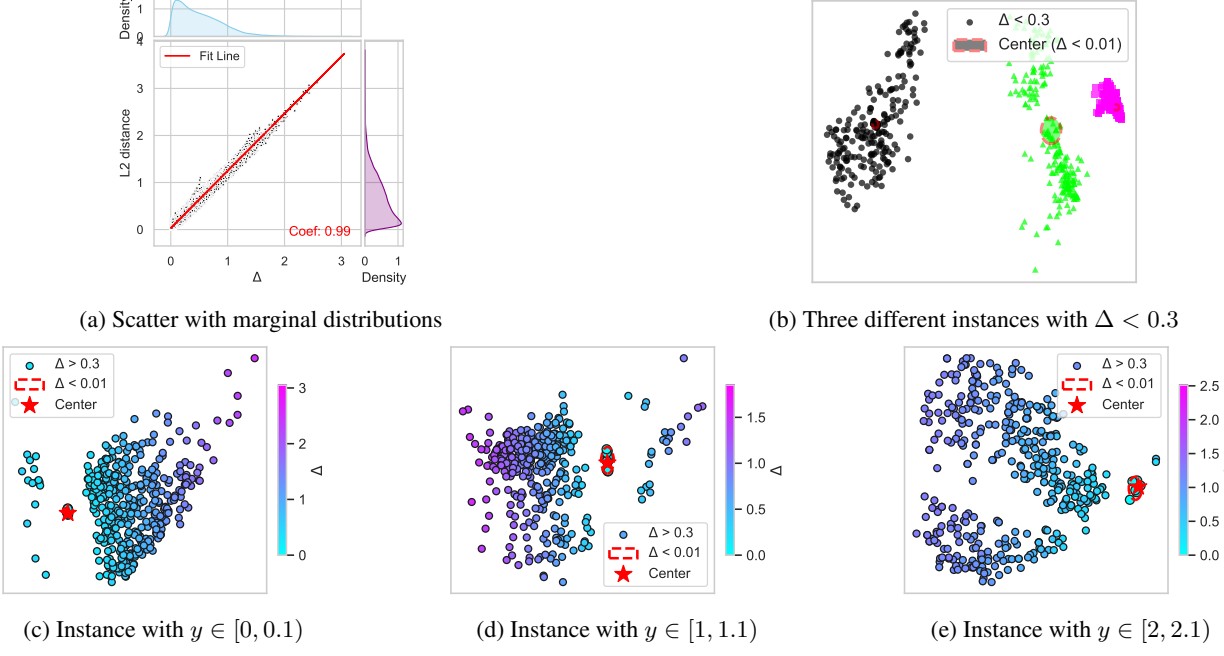

(a) Scatter with marginal distributions

(b) Three different instances with $\Delta < 0.3$

(c) Instance with $y \in [0, 0.1)$

(d) Instance with $y \in [1, 1.1)$

(e) Instance with $y \in [2, 2.1)$

*Figure 5.* ProWTP on Wechat.

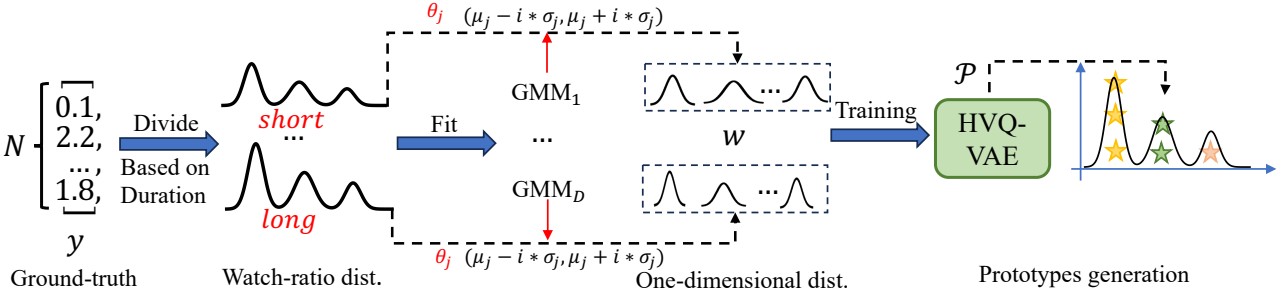

*Figure 6.* The details of Pre-processing and Prototype generations.

3. **Random Sampling.** For each GMM, $C$ sets of means, variances and weights $\{(\mu_j, \sigma_j, \theta_j)\}_{j=1}^{C}$ are obtained. The sampling process is as follows:

- Data for each peak is sampled randomly to form a distribution $w = (y_1, y_2, \ldots, y_n)$ of length $n$, with the sampling range defined as $[\mu - i \cdot \sigma, \mu + i \cdot \sigma]$, where $i$ follows a Gaussian distribution $N(\mu', \sigma')$.

- To ensure that the overall sampled distribution matches the original distribution, the number of samples for each peak is determined by the weights $\theta$. Specifically, when generating $L$ near-Gaussian distributions from the current multi-modal distribution, the allocation of $L$ is governed by $\theta$, where the number of samples generated for each peak is $\lfloor \theta \cdot L \rfloor$.

This sampling strategy significantly simplifies subsequent learning while adhering to the semantic of Prototypes, where each Prototype represents the center of a peak.

4. **Generating Credible Prototypes.** For each bucket, $L$ distributions $w$ are sampled, resulting in $D \times L$ distributions. These are fed into the HVQ-VAE for training, and the codebook weights from HVQ-VAE are considered as Prototypes.

In Fig. 7 and 8, we present a comparison of the overall distribution of 100 sampled $w$ values (orange) and the original watch-ratio distribution (blue) across different durations on the WeChat and our Short-video datasets, respectively. The red dashed lines indicate the means of the GMM. It can be observed that the distributions exhibit typical multimodal characteristics, and the sampled distributions successfully preserve the original distribution's shape and features.

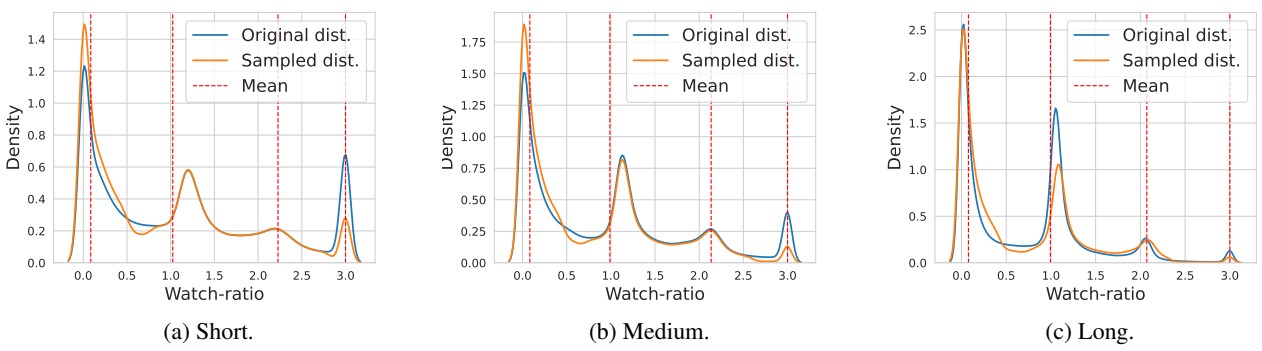

(a) Short.  (b) Medium.  (c) Long.

*Figure 7.* Comparison of Sampled Distribution and Original Watch-Ratio Distribution on WeChat.

### A.3. The Relationship Between Multimodal Distributions, Prototypes, and User Behavior.

The watch-ratio distribution exhibits distinct multi-modal characteristics, reflecting different user behavior patterns during video consumption: "scroll" (the first peak) indicates that users skim past the video after watching the cover for about 1 second, showing a lack of interest; "like" (the second peak) represents users who watch most of the video and show moderate interest; and "very like" (subsequent peaks) suggests users who are highly engaged with the content and may even

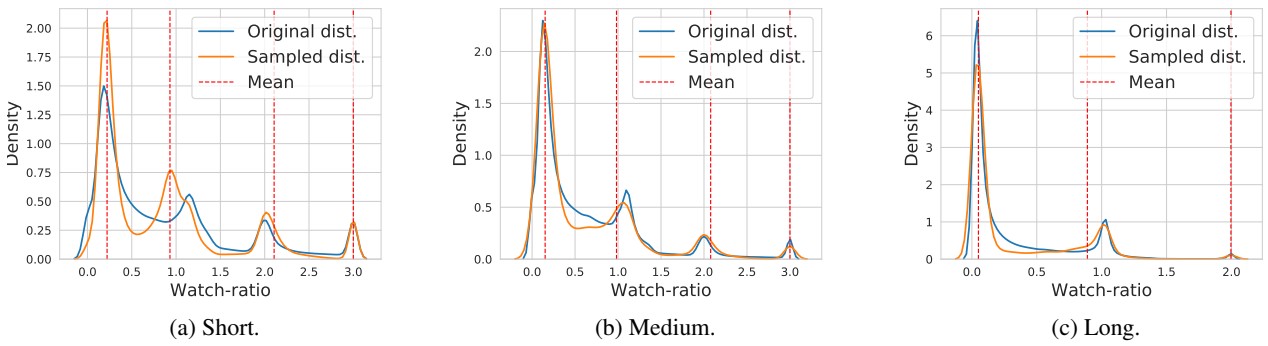

(a) Short.  (b) Medium.  (c) Long.

*Figure 8.* Comparison of Sampled Distribution and Original Watch-Ratio Distribution on our Short-video.

re-watch it multiple times. This clustering behavior helps watch-time prediction models quickly identify specific intervals, thereby reducing prediction errors.

However, multi-modal distributions are typically long one-dimensional sequences, making direct modeling challenging for capturing behavior patterns effectively. Prototype learning addresses this issue by dividing the multi-modal distribution into several sub-distributions and generating multiple semantic centers in high-dimensional space for each sub-distribution. This approach significantly simplifies computation and learning complexity, breaking down the complex multi-modal distribution into more manageable local structures. Consequently, Prototype enables watch-time prediction models to more accurately capture the characteristics of different user behavior patterns, improving global prediction performance of duration distributions and effectively supporting recommendation systems in WTP tasks.

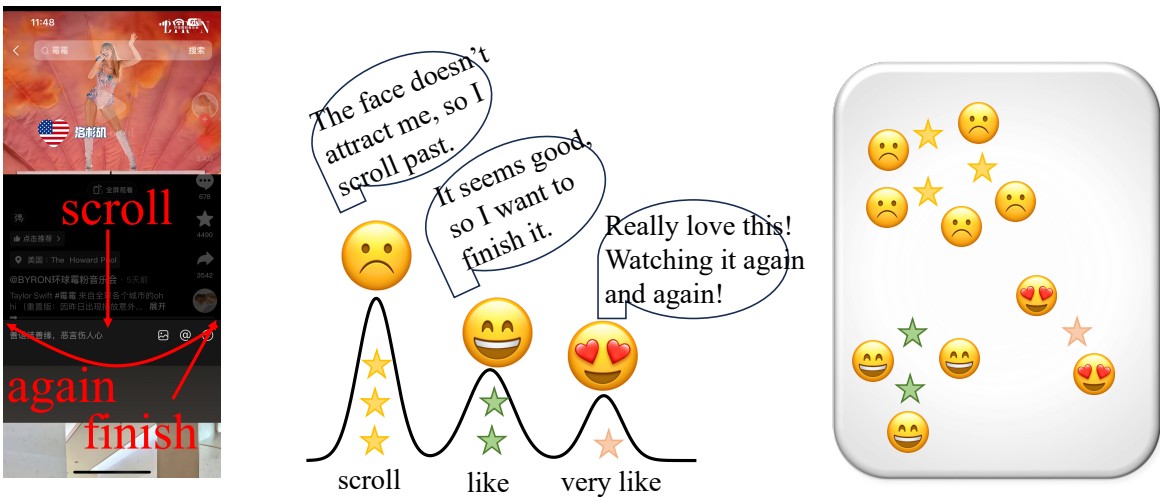

*Figure 9.* The Relationship Between Multimodal Distributions, Prototypes, and User Behavior.

### A.4. Computational Complexity Discussion

**Stage I: prototype generations.** The computational complexity of HVQ-VAE primarily arises from the cluster selection and prototype selection. For a single sample, the cluster selection involves computing the attention-weighted cluster centers, with a time complexity of $O(C \cdot K + C)$, where $C$ is the number of clusters, $K$ is the number of prototypes per cluster. Within the selected cluster, prototype selection further incurs a complexity of $O(K)$. Overall, the time complexity for a single sample is $O(C \cdot K + C + K)$, and the space complexity is dominated by the static storage of the codebook, which is $O(C \cdot K \cdot d)$, and $d$ is the prototype vector dimension.

Importantly, HVQ-VAE is completely independent, and its spatio-temporal complexity does not affect the training and inference time of ProWTP. When the distribution of watch ratios is sufficiently large, the resulting prototype distribution is stable. Furthermore, we observed that for a well-established video recommendation APP, the watch-ratio distribution remains largely unchanged and consistent across multiple months.

As shown in Figure 10, we randomly sampled 200,000 users from our APP (a short-video platform) and extracted their historical behavior on the 1st day of each month from January to November 2024. The data were divided into $D = 15$ buckets based on video duration. We then computed the Wasserstein Distance between the watch-ratio probability density distributions of each month and November, as well as the Kolmogorov-Smirnov test with $p < 0.05$ between their cumulative empirical distributions. The results indicated no significant distribution shifts across multiple months.

Even in extreme scenarios where user behavior undergoes notable adjustments, we only need to resample the watch-ratio distributions for each duration buckets, perform offline retraining, and update the weights of ProWTP. This process incurs minimal computational overhead.

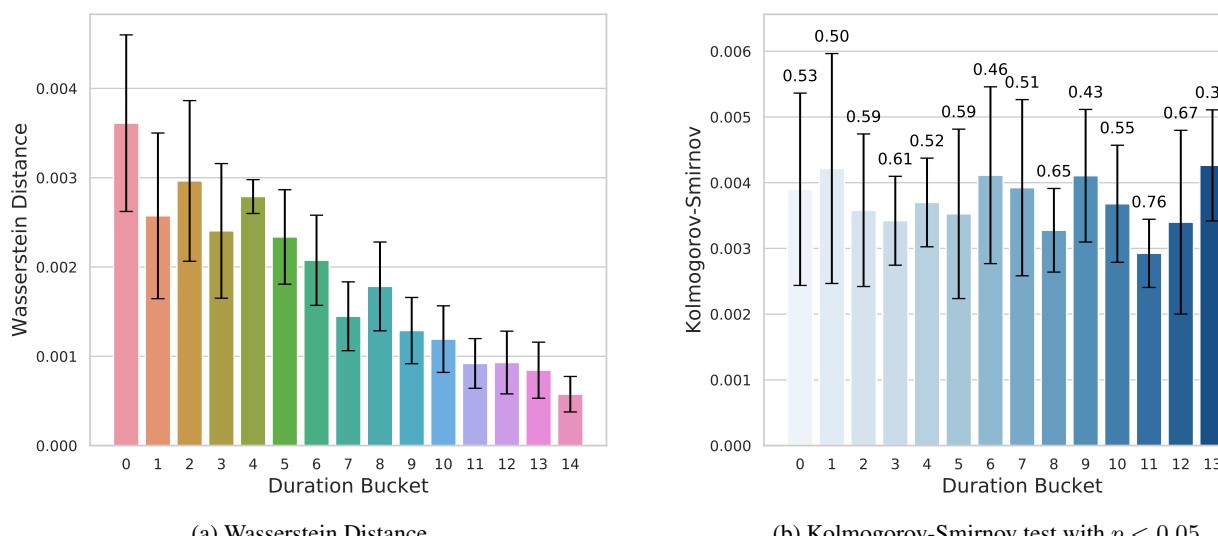

(a) Wasserstein Distance.  (b) Kolmogorov-Smirnov test with $p < 0.05$

*Figure 10.* Mean Differences in Watch-ratio Distributions Between November and January-October.

**Stage II: Distribution alignment.** ProWTP is a model-agnostic method that adds only an additional prototype layer compared to the baseline, resulting in a space complexity of $O(CKd)$. During the inference phase, the OT module is removed, and the final value is computed as a linear combination of similarities to each prototype, which is then input into the regressor. The time complexity of this process is $O(CK)$, where $C$ and $K$ are small constants, ensuring that the time overhead remains negligible. The training time complexity comes from four parts. OT optimization operates on a transportation matrix of size $n_b \times CK$, where $n_b$ is the mini-batch size and $CK$ is the number of prototypes, with a complexity of $O(I \cdot n_b \cdot CK)$ for $I$ iterations. The calibration loss, which computes softmax and cross-entropy for each sample across all prototypes, has a complexity of $O(n_b \cdot CK)$. The compact loss, which encourages tighter clustering of instance representations under the same prototype, involves sampling $20\%$ of the instances and computing pairwise similarities, with a complexity of $O(0.04 \cdot CK \cdot |\mathcal{S}_k^+|^2)$. Additionally, the prototype-weighted prediction calculation incurs an additional $O(n_b \cdot CK)$. Thus, the overall training time complexity for a batch is $O(I \cdot n_b \cdot CK + 2 \cdot n_b \cdot CK + 0.04 \cdot CK \cdot |\mathcal{S}_k^+|^2)$.

We trained on WeChat data with a batch size of 512 using an RTX 4090 GPU. Tab. 5 compares the training time per batch for ProWTP under different sampling frequencies and the corresponding changes in RMSE, along with the inference efficiency of different models. It can be observed that ProWTP's inference efficiency does not significantly increase compared to the baseline. However, as the sampling ratio increases, the training time for ProWTP grows noticeably, while the performance improvement shows diminishing marginal returns.

*Table 5.* Time cost (s) per batch of different models on Wechat.

| Model | TR | D2Q | ProWTP | | | | | |
|---|---|---|---|---|---|---|---|---|
| Sample ratio | - | - | 0% | 10% | 20% | 30% | 50% | 100% |
| Train cost | 0.011 | 0.013 | 0.049 | 0.058 | 0.061 | 0.075 | 0.092 | 0.121 |
| RMSE | 30.39 | 29.12 | 29.38 | 28.91 | 28.47 | 28.22 | 28.05 | 28.04 |
| Infer cost | 0.003 | 0.003 | 0.004 | | | | | |

## A.5. Datasets.

**1) Wechat**: This dataset was adopted in WeChat Big Data Challenge[1], which records the behavior of users on short videos in two weeks. We divide the duration into $D = 5$ buckets. The user_id, device_id, video_id, author_id, duration_level and multi-model content feature vectors are used as our feature inputs.

**2) Kuairand-Pure**: Constructed from the recommendation logs of the video-sharing mobile app, Kuaishou (Gao et al., 2022), the dataset contains millions of intervened interactions about 27,285 users and 7,551 items in 4 weeks. Similarly, we discretize the duration into $D = 5$ buckets in this dataset, and the user_id, video_id, tab, music_id, author_id, duration_level and user_active_degree will serve as input features in our experiments.

**3) Short-video**: We collected behavioral logs of 200,000 active users from a short-video platform on November 1, 2024. The data was divided into $D = 15$ buckets based on video duration. In our experiments, we used the following features as inputs: user_id, video_id, tag_id, author_id, and duration_level.

*Table 6.* Statistical Information of datasets.

| Data | #user | #video | #interaction | #duration |
|---|---|---|---|---|
| WeChat | 20,000 | 96,428 | 7,210,290 | 5 |
| Kuairand-Pure | 27,285 | 7,551 | 1,231,181 | 5 |
| Short-video | 200,000 | 4,832,885 | 30,000,000 | 15 |

## A.6. Baseline Details.

To evaluate the effectiveness of our proposed method, we compare it with the following methods that are pivotal in leveraging Watch-time prediction:

- **TR (Traditional Regression)**: This method adopts a straightforward regression approach, using watch time as the label. It is trained to minimize the Mean Squared Error (MSE).

- **WLR (Weighted Logistic Regression)** (Covington et al., 2016): As implemented in YouTube's system, this method learns a logistic regression model, reweighted by watch times, and uses the learned odds to estimate watch time during prediction.

- **OR (Ordinal Regression)** (Crammer & Singer, 2001): This method, based on ordinal regression techniques, emphasizes the relative order of watch times, fitting the data to predict categorical watch time levels.

- **D2Q (Duration-Deconfounded Quantile)** (Zhan et al., 2022): Representing a state-of-the-art approach in watch time

---

[1]https://algo.weixin.qq.com/2021/problem-description

prediction, this model addresses duration bias through backdoor adjustment and fits duration-dependent quantiles of watch time using MSE.

- **TPM (Tree-based Progressive Model)** (Lin et al., 2023): This approach uses a tree-structured series of classification tasks, considering ordinal ranks and prediction variance, and incorporates backdoor adjustment to mitigate bias, offering a nuanced and comprehensive approach to enhancing watch time prediction in video recommender systems.

- **DVR (Debiased Video Recommendation)** (Zheng et al., 2022): This methods provides unbiased recommendation of micro-videos with varying duration, and learn unbiased user preferences via adversarial learning.

- **CWM (Counterfactual Watch Model)** (Zhao et al., 2024): This methods proposes to use counterfactual reasoning to mitigate duration bias.

### A.7. Metrics.

**Root Mean Square Error (RMSE)**. This metric measures the average magnitude of errors between generated values and actual values, which is formulated as:

$$\text{RMSE} = \sqrt{\frac{1}{n} \sum_{i=1}^{n} (y_i - \hat{y}_i)^2}, \tag{25}$$

where $y_i$ is the actual value of the $i$-th sample and $\hat{y}_i$ is the predicted value.

**Mean Absolute Error (MAE)**. This metric is used to evaluate the average discrepancy between generated and real data; the calculation is as follows:

$$\text{MAE} = \frac{1}{n} \sum_{i=1}^{n} |y_i - \hat{y}_i|. \tag{26}$$

**XAUC** (Zhan et al., 2022). This is an extension of the standard AUC, applied to continuous values. Given a pair of samples $(i, j)$, if the predicted watch-time values $\hat{y}_i$ and $\hat{y}_j$ are in the same order as their true values $y_i$ and $y_j$, the score is 1; otherwise, the score is 0. We uniformly sample such pairs from the test set, and the XAUC is computed as the average score over all pairs. The formal definition is:

$$\text{XAUC} = \frac{1}{|\mathcal{S}|} \sum_{(i,j)\in\mathcal{S}} \mathbb{I}\left[(\hat{y}_i > \hat{y}_j) = (y_i > y_j)\right], \tag{27}$$

where $\mathcal{S}$ represents the set of all sampled pairs, and $\mathbb{I}(\cdot)$ is the indicator function, which returns 1 if the predicted order matches the true order, and 0 otherwise. XAUC intuitively measures how well the ranking induced by the predicted watch times aligns with the true ranking. A higher XAUC indicates better model performance.

**XGAUC** (Zhan et al., 2022). This is a weighted version of XAUC. It computes XAUC for each user individually, and then averages the XAUC values with weights proportional to the sample size of each user. The formal definition is:

$$\text{XGAUC} = \frac{\sum_u N_u \cdot \text{XAUC}_u}{\sum_u N_u}, \tag{28}$$

where $u$ represents a user, $N_u$ is the number of samples for user $u$, $\text{XAUC}_u$ is the XAUC score for user $u$. XGAUC measures the overall ranking consistency across users, with the weight adjusted based on the number of samples per user. A higher XGAUC indicates better model performance across users.

In WTP tasks, MAE and RMSE are used to measure how close the predicted watch times are to the actual values, focusing on the accuracy of the predictions. XAUC and XGAUC, on the other hand, evaluate how well the predicted rankings of watch times match the true rankings, emphasizing the importance of the order of predictions. Both metrics are crucial: accurate predictions (measured by MAE and RMSE) ensure precision, while correct rankings (measured by XAUC and XGAUC) are essential for delivering relevant recommendations. In recommendation systems, maintaining the correct ranking is often as important, if not more so, than predicting the exact values, making both aspects vital for optimizing user satisfaction and overall model performance.

**A.8. The derivation of $\mathcal{L}_{HVQ-VAE}$.**

The loss function $\mathcal{L}_{HVQ-VAE}$ is designed to optimize both the encoder and decoder networks, while preserving the discrete nature of the latent space.

$$\mathcal{L}_{\text{HVQ-VAE}} = \left\| \mathbf{w} - D\big(E(\mathbf{w}) + \text{sg}[\mathbf{z} - E(\mathbf{w})]\big) \right\|_2^2 \tag{29}$$
$$+ \left\| \text{sg}[E(\mathbf{w})] - \mathbf{z} \right\|_2^2 + \beta \left\| E(\mathbf{w}) - \text{sg}[\mathbf{z}] \right\|_2^2.$$

This function consists of three key components, explained as follows:

**Reconstruction loss:**

$$||\mathbf{w} - D(E(\mathbf{w}) + \text{sg}[\mathbf{z} - E(\mathbf{w})])||_2^2 \tag{30}$$

This part measures the squared Euclidean distance between the decoder output $D(\cdot)$ and the original input $\mathbf{w}$, assessing the model's ability to reconstruct the data. Here, $E(\mathbf{w})$ represents the encoder output of the input $\mathbf{w}$, and $\mathbf{z}$ is the nearest prototype vector. The stop-gradient operation $\text{sg}[\cdot]$ prevents gradients from passing through, ensuring that the codebook is only updated through the second term. During forward propagation (when calculating the loss), this simplifies to $D(E(\mathbf{w}) + \mathbf{z} - E(\mathbf{w})) = D(\mathbf{z})$, and during backpropagation (when calculating the gradients), since $\mathbf{z} - E(\mathbf{w})$ provides no gradients, it also simplifies to $D(E(\mathbf{z}))$.

**Quantization Loss:**

$$||\text{sg}[E(\mathbf{w})] - \mathbf{z}||_2^2 \tag{31}$$

This loss encourages the prototype vector $\mathbf{z}$ to move closer to the encoder output $E(\mathbf{w})$. The stop-gradient operation is applied to $E(\mathbf{w})$ to prevent gradients from propagating through this term to the encoder, thus only updating the codebook.

**Commitment Loss:**

$$\beta||E(\mathbf{w}) - \text{sg}[\mathbf{z}]||_2^2 \tag{32}$$

This term encourages the encoder output $E(\mathbf{w})$ to commit to the chosen codebook vector $\mathbf{z}$. The weight factor $\beta$ adjusts the importance of this loss relative to the other components. By increasing the encoder's commitment to its quantized representation, this term improves the model's stability and efficiency.

**A.9. Results on different duration buckets.**

**A.10. Results on different duration buckets.**

Table 7. Results on different duration buckets.

| Duration bucket | Wechat | | | | KuaiRand-Pure | | | |
|---|---|---|---|---|---|---|---|---|
| | RMSE | MAE | XAUC | XGAUC | RMSE | MAE | XAUC | XGAUC |
| 0 | 8.57 | 6.87 | 0.6118 | 0.5273 | 7.82 | 5.57 | 0.6922 | 0.6365 |
| 1 | 13.13 | 10.53 | 0.6084 | 0.5334 | 16.40 | 12.37 | 0.6689 | 0.6085 |
| 2 | 20.15 | 16.19 | 0.6088 | 0.5289 | 28.79 | 21.45 | 0.6941 | 0.6245 |
| 3 | 31.95 | 26.00 | 0.5930 | 0.5261 | 43.62 | 32.03 | 0.6786 | 0.6167 |
| 4 | 48.97 | 40.34 | 0.5795 | 0.5203 | 69.09 | 48.40 | 0.6614 | 0.6153 |

**A.11. Training Loss.**

$$\mathcal{L} = \mathcal{L}_{task} + \mathcal{L}_{assign} + \beta * \mathcal{L}_{compact}, \tag{33}$$

where $\beta$ is the hyper-parameter ranged from $(0.0, 0.2]$.

**A.12. Why OT?**

Assuming the instance representation is $\mathbf{h}_i$ and the prototype set is $\{\mathbf{p}_k\}_{k=1}^{C*K}$, the weight between $\mathbf{h}_i$ and each prototype $\mathbf{p}_i$ is defined as:

$$\alpha_{i,k} = \frac{\exp\left(\mathbf{h}_i^T * \mathbf{p}_k / \tau\right)}{\sum_{j=1}^{C*K} \exp\left(\mathbf{h}_i^T * \mathbf{p}_j / \tau\right)}. \tag{34}$$

We consider the three different alignment methods:

- SUOT calculates a transport matrix $\mathbf{T}$ based on the relationship between prototypes and instances, and uses $t_{i,k} \in \mathbf{T}$ to guide the learning of $\alpha$. This approach considers global distribution alignment, offering strong robustness and interpretability.:

$$\mathcal{L}_{assign} = -\frac{1}{n_b} \sum_{i=1}^{n_b} \sum_{k=1}^{C*K} t_{i,k} \log \alpha_{i,k}. \tag{35}$$

- L2 distance directly aligns two representations, focusing on point-wise alignment without considering the global distribution. This makes it susceptible to the influence of outliers.:

$$\mathcal{L}_{assign} = ||\mathbf{h}_i - \sum_{k=1}^{C*K} \alpha_{i,k} * \mathbf{p}_k||_2. \tag{36}$$

- w/o alignment directly uses the linear combination $\sum_{k=1}^{C*K} \alpha * \mathbf{p}_k$ for prediction without $\mathcal{L}_{assign}$.

Tab. 4 and 8 compare the results of different alignment methods, showing that SUOT achieves the best performance, which demonstrates the effectiveness of OT-based alignment. Fig. 11 provides a case study where we visualize the weight matrix $\alpha$ of a batch ($n_b = 512$, $C * K = 80$) from the WeChat dataset. It can be observed that the $\alpha$ learned by OT alignment maintains the same sparsity as the transport matrix $\mathbf{T}$. In contrast, the $\alpha$ from other methods is very dense, treating the prototypes as mere representation anchors to enhance the overall representation, while ignoring whether instances should actually match their corresponding prototypes.

*Table 8.* Different distribution alignment methods.

| Distribution alignment | Wechat | | | | KuaiRand-Pure | | | |
|---|---|---|---|---|---|---|---|---|
| | RMSE | MAE | XAUC | XGAUC | RMSE | MAE | XAUC | XGAUC |
| SUOT | **28.47** | **19.84** | **0.6180** | **0.5727** | **40.44** | **24.33** | **0.7288** | **0.7045** |
| L2 distance | 29.37 | 20.35 | 0.6129 | 0.5683 | 41.28 | 24.91 | 0.7208 | 0.7006 |
| w/o alignment | 29.90 | 20.89 | 0.6108 | 0.5665 | 42.00 | 25.50 | 0.7185 | 0.6980 |

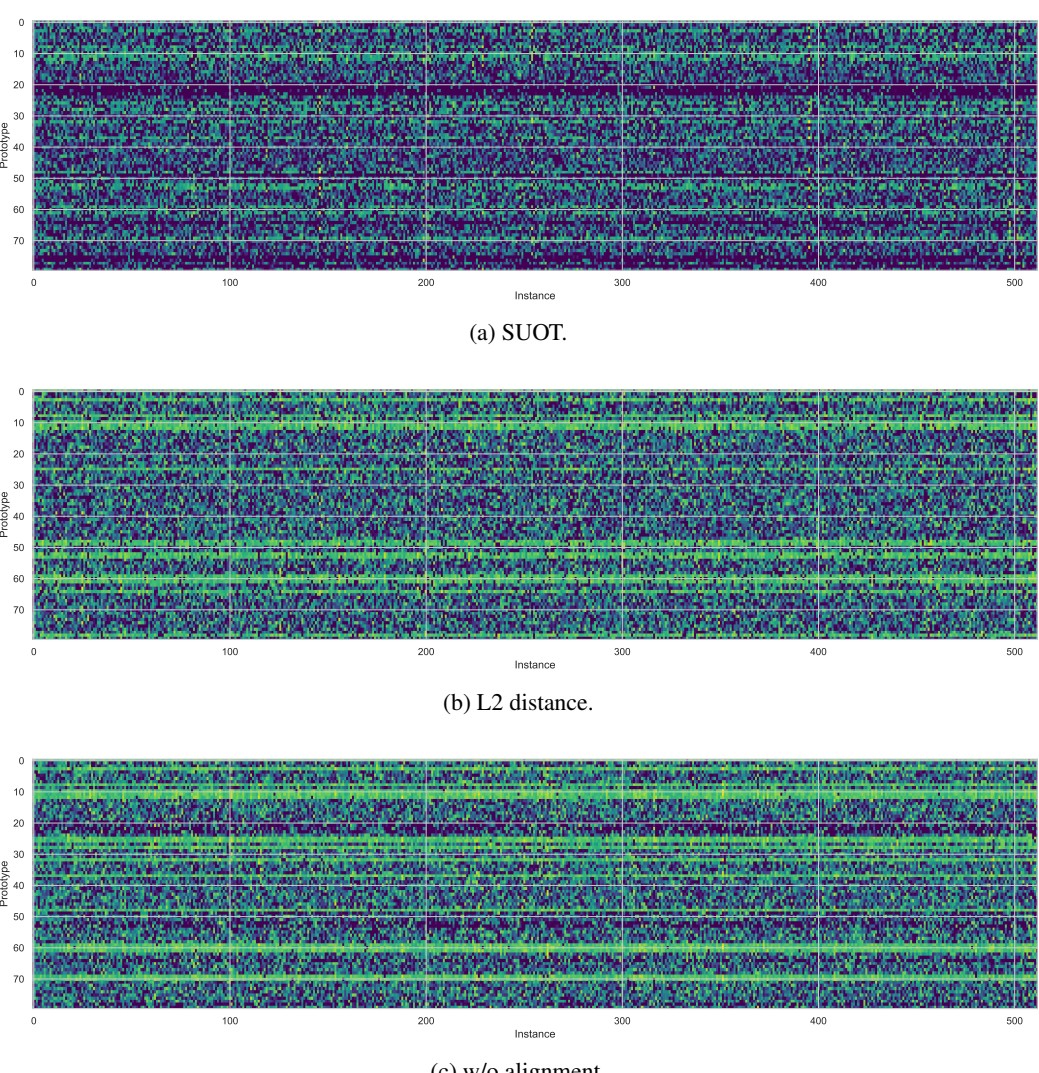

(a) SUOT.

(b) L2 distance.

(c) w/o alignment.

*Figure 11.* A case study on the weights $\alpha$ for different alignment methods.

