# OpenReview forum: "Calibrating Video Watch-time Predictions with Credible Prototype Alignment"
_ICML.cc/2025/Conference — ICML 2025 poster_

### Official Review · Reviewer_KkGb · 2025-03-01

**Overall Recommendation:** 1

**Summary:**

The paper proposes ProWTP, a two-stage approach designed to enhance watch-time prediction in video recommender systems by integrating prototype learning with optimal transport (OT). In the first stage, ProWTP employs a hierarchical vector quantized variational autoencoder (HVQ-VAE) to transform continuous watch ratio labels into discrete prototypes. In the second stage, optimal transport is used to align the distribution of these labels with the instance representation distribution, thereby boosting prediction accuracy. Extensive offline experiments conducted on three datasets show that ProWTP outperforms existing methods.

#########
Added after rebuttal: I checked the authors' rebuttal, and gave a detailed response. In summary, the authors' feedback doesn't resolve my concerns, so I maintain my original ranking of the paper.

**Claims And Evidence:**

No. This work is problematic in at least the following aspects:

1. Unclear Motivation: The submission does not adequately justify the motivation of the proposed methods (e.g. D2Q for debiasing, TPM for reducing classification difficulty). The claim, "However, those methods struggle to consistently maintain high predictive accuracy across different models," was made without detailed explanation or evidence. The authors should strengthen the motivation of their approach.

2. Overgeneralized Claim: The assertion that the method is "suitable for any deep recommendation model" (in line 52) is questionable. In scenarios where data lacks multimodal distributions, the model’s robustness is uncertain, and no evidence addresses this limitation.

3. Insufficient Comparison: The authors claim, "Different from traditional prototype learning, ProWTP generates prototype vectors from label distributions, providing models with more precise and credible calibration references." (in line 81) While metrics are provided in tables, the superiority over simple clustering-based contrastive learning (which could also handle multimodal predictions) is not convincingly demonstrated. Visualizations comparing ProWTP with clustering, beyond the TR vs. ProWTP in the appendix, would enhance credibility.

4. Efficiency Concerns: The use of optimal transport likely incurs high computational cost, yet the efficiency of the proposed two-stage online approach is not adequately addressed, raising doubts about its practicality.

5. Lack of Literature Review: The paper lacks a thorough review of prior watch time prediction research and omits key references. Notably, the claim, "We investigate the multimodal distribution properties of watch-ratio across different video duration buckets for the first time,"(in line 105) is inaccurate, as the bimodal distribution of KuaiRand was previously studied in Zhao et al., 2023.

[1] Zhao et al., 2023, Uncovering User Interest from Biased and Noised Watch Time in Video Recommendation

**Essential References Not Discussed:**

1. The paper does not provide a thorough review of prior research on watch time prediction and omits several important references. Specifically, the claim in line 105, "We investigate the multimodal distribution properties of watch-ratio across different video duration buckets for the first time," is inaccurate. The bimodal distribution of KuaiRand was previously explored in [1], undermining the novelty asserted by the authors.

2. Additionally, the paper lacks comparisons with key prior works, including:
CREAD[2], SWAT[3], GR[4].

3. The reported improvements over baselines on the two datasets appear modest, warranting a statistical significance test to validate their impact. To strengthen the evaluation, the authors should compare their method against these approaches and test performance on additional benchmarks such as KuaiRec and CIKM datasets.

4. The paper lacks discussion on interpreting prototypes and their connection to user behavior, which could hinder insight into the model’s predictions.
[1] Zhao et al., 2023, Uncovering User Interest from Biased and Noised Watch Time in Video Recommendation
[2] CREAD: A Classification-Restoration Framework with Error Adaptive Discretization for Watch Time Prediction in Video Recommender Systems
[3] SWaT: Statistical Modeling of Video Watch Time through User Behavior Analysis
[4] Generative Regression Based Watch Time Prediction for Video Recommendation

**Experimental Designs Or Analyses:**

Yes, I checked the soundness/validity of any experimental designs or analyses. I have the following concerns:
1. Visualization in Appendix: While the appendix visualizes a comparison between TR and ProWTP, it lacks a comparison with clustering-based methods. Including such a comparison (e.g., ProWTP vs. clustering) would strengthen the evidence for ProWTP’s superiority, as clustering could also address multimodal distributions, making the current visualization less convincing.

2. Rationale for Optimal Transport: It is unclear why aligning the feature space with prototype vectors via optimal transport improves watch time prediction. Simpler alternatives (e.g., cosine similarity or k-means) might achieve similar alignment. The authors should justify the choice of optimal transport and explain its specific benefits for the prediction task.

3. Efficiency Trade-offs: The trade-off between performance gains and computational efficiency is concerning. Although the authors sample 20% of the data for contrastive learning, the per-sample cost of optimal transport appears substantial. Additionally, the deployment strategy is ambiguous: Are prototypes fixed during online inference? If so, how does the model address distribution shifts, and what are the implications for robustness?

4. Robustness Across Scenarios: The claim "suitable for any deep recommendation model" is tested only on multimodal data. The experimental design lacks validation on scenarios without multimodal distributions (e.g., unimodal data), leaving the model’s generalizability and robustness unverified.

**Methods And Evaluation Criteria:**

Yes.

**Other Comments Or Suggestions:**

No.

**Other Strengths And Weaknesses:**

Figure 1b fails to specify the method used or which data components are analyzed to demonstrate "Representation confusion." In Figure 2, the boundaries and logic between modules are poorly defined and highly confusing. Overall, the quality of the figures is substandard, detracting from the paper’s readability and professionalism.

**Questions For Authors:**

Please refer to the problems above for detailed weaknesses.

**Relation To Broader Scientific Literature:**

The contributions of the paper were built upon and extending several strands of work in the scientific literature:
1.Prototype Learning and Representation Discretization: Based on methods like VQ-VAE, the paper extends unsupervised learning and clustering techniques by converting continuous labels into discrete prototype.

2.Optimal Transport for Distribution Alignment: Optimal transport has been widely used to address distribution mismatches. Here, it is employed to align the label-based prototype distribution with instance representations for for video watch-time prediction

**Theoretical Claims:**

Yes.

---

> ### Author Rebuttal · Authors · 2025-03-31
>
> Dear Reviewer KkGb,
>
> We appreciate your valuable questions and suggestions. We summarize your concerns below and provide responses.
>
> > **Q1: The motivation is clear and explicit.**
>
> We mention that existing WTP models struggle to achieve high accuracy. We believe the main reason lies in instance representation confusion, which we explain from two perspectives:
>
> 1. Mathematical Explanation
>
>  let the instance representation of a sample $(x,y)$ be $f(x)$, with its ideal center being $ \mu_y = \mathbb{E}[f(x) \mid y]$, where $y$ is the ground-truth. The degree of instance representation confusion is defined as the distance between the instance representation and the ideal center $ d(f(x), \mu_y) = \|f(x) - \mu_y\|.$ Then, the model's prediction error $ \Delta_x = |y - \hat{y}|$ is predominantly correlated with the degree of instance representation confusion $d(f(x), \mu_y)$.
>
> 2. Model Performance
>
> In existing WTP models, the instance representation space is often disorganized, with instance representations of different data types failing to form well-defined clusters, as shown in Figure4. Also, we provide additional results of different duration buckets in **Reviewer WSao Q3**. ProWTP's performance is even better in medium and long videos with large prediction errors, which further illustrates that ProWTP optimizes the distance between instance representations and prototypes to achieve more accurate predictions.
>
> We believe that ProWTP improves prediction accuracy primarily by alleviating instance representation confusion and directly incorporating label distributions into the model, thereby providing additional information gain. So the motivation is clear and explicit.
>
> > **Q2: Suitable for any deep recommendation model**
>
> This statement wants to show that ProWTP is a model-independent scheme, i.e., $f$ in $f(x)$ can be replaced by any depth model (i.e. DCN), and there is no overclaim.
>
> > **Q3: Comparison with traditional clustering**
>
> In Table 3, we compare Kmeans' scheme for generating prototype. As for the clustering-based contrastive learning scheme, we have not searched relevant literature in WTP at this time.
>
> > **Q4: Efficiency**
>
> **A.4.** has discussed this problem in details,  OT is only present during training, OT is removed during testing, and online inference time complexity is linear O(CK).
>
> > **Q5: lack of references**
>
> **Its not TRUE. line 096 in Related work part, [1] proposes bimodal distribution, which we have already cited**. In our work, we focus on the multi-modal distribution characteristics of videos. Previous studies have overlooked the distribution of watch-time beyond the video duration, and we are the first to propose focusing on the multimodal characteristics rather than just bimodal distribution. Importantly, we need to emphasize that the main contribution of our paper is the transformation of label distributions into prototypes that provide a credible reference for model calibrations.
>
> > **Q6: Visualization**
>
> Thanks for your suggestion, we will add the use of kmeans-generated prototypes for ProWTP visualization subsequently.
>
> > **Q7: Rationale for OT**
>
> **In Appendix A.12, we compare three different alignment methods: OT, L2 distance, and no alignment.** We present their results and conduct a case study. The OT alignment method performs global alignment rather than aligning each sample independently. Compared to other alignments, OT introduces structured information. This approach not only achieves better results but also produces sparser weights between instances and prototypes.
>
> > **Q8: Efficiency Trade-off**
>
> **A.4.** has discussed this problem in details.  We tested the wr distribution across months and its distribution does not change significantly, so it is reasonable that prototype is naturally fixed. **If it changes in extreme cases, we just need to re-pull the data for training, as is always done in industry.**
>
> > **Q9: Robustness Across Scenario**
>
> This scenario does not occur in the WTP task, because according to our platform data analysis (its DAU is 400 millions), **the users' watch-ratio always shows a multi-modal distribution. This is inherent in the WTP.**
>
> > **Q10: only two datasets**
>
> **ITS NOT TRUE**.  In our paper it is **three datasets** and the results are reported averaged over five runs.
>
> > **Q11. The interpretability of prototypes and their association with user behavior .**
>
> It has been discussed in **A.3** in details.  Due to space constraints, you can refer to A.3.
>
> > **Q12: More baselines**
>
> | Wechat | CREAD | SWAT  | ProWTP |
> |--------|-------|-------|--------|
> | RMSE   | 28.93 | 29.31 | 28.47  |
>
> The time of GR in arxiv is Sat, 28 Dec 2024, which is a same-time work and can be disregarded.
>
> > **Q13: Figures modification**
>
>  Due to space constraints, you can refer to Reviewer WSao Q1.
>
> We sincerely thank you again for your feedback and hope that our responses can change your mind. We look forward to your reply and further discussion.

---

### Official Review · Reviewer_3cw2 · 2025-03-11

**Overall Recommendation:** 4

**Summary:**

This paper focuses on the watch-time prediction problem in video recommender systems. It employs a two-stage framework: (1) using a hierarchical vector quantized variational autoencoder to generate credible prototypes from watch-ratio distributions; and (2) leveraging semi-relaxed unbalanced optimal transport to align samples with those prototypes. The goal is to alleviate “instance representation confusion” that leads to prediction errors. Appendix gives the corresponding mathematical proof and statistical analysis that justifies the motivation of the paper. This paper provides offline experiments and online A/B tests to verify the effectiveness of the method in real scenarios. Overall, this paper has a very fresh perspective, is undeniably innovative, and presents interesting ideas and scenarios.

**Claims And Evidence:**

The authors argue that conventional methods fail to fully exploit the multi-modal nature of watch-ratio under different video-duration buckets and often overlook the confusion that arises in deep recommendation model representations. By generating credible prototypes on the label side and performing distribution alignment, their method substantially reduces representation offset, thereby lowering prediction errors. Empirical evidence is provided by experiments on multiple real-world industrial datasets, where the proposed approach outperforms a wide range of baselines, demonstrating its effectiveness.

**Essential References Not Discussed:**

I do not find such related works.

**Experimental Designs Or Analyses:**

The authors conduct experiments on multiple real-world datasets, comparing with various watch-time prediction baselines (like D2Q, TPM) and debiasing models (like DVR, CWM). They also perform ablation studies to assess how removing HVQ-VAE, SUOT, or the assignment loss affects performance. Additional visualizations illustrate how the representation space becomes significantly less confused once samples are attracted to their respective prototypes.

**Methods And Evaluation Criteria:**

The papers proposed two-stage approach involves:
1. Using HVQ-VAE to quantize the watch-ratio distribution into prototypes for model calibration;
2. Employing SUOT to align sample representations with these prototypes, along with an assignment loss and a compactness loss to further ensure samples cluster around their respective prototypes.
Evaluation uses RMSE and MAE to measure regression errors, and XAUC and XGAUC to assess ranking performance.

**Other Comments Or Suggestions:**

I have no other comments and suggestions.

**Other Strengths And Weaknesses:**

Strengths:
1. The paper proposes a novel and interesting two-stage method that stands out by quantizing label distributions themselves—rather than relying on solely sample-based embeddings—to derive meaningful prototypes for instance representation calibrations.
2. A Well-Motivated paper.  By focusing on “instance representation confusion,” the authors provide both an intuitive and a formal rationale for aligning sample representations with label-derived prototypes
3. The paper presents experiments on multiple real-world datasets as well as online ab test, systematically comparing the proposed method with a range of established baselines.
4. In addition to empirically validating their approach, the authors incorporate a rigorous theoretical analysis, particularly in Appendix A.1, where they link prediction error to the distance from an “ideal center.” This mathematical proof underpins why the prototype alignment mechanism effectively curbs representation confusion and improves watch-time prediction accuracy.

Weakness:
1. The proposed two-stage framework may introduce additional modeling complexity compared to simple direct regression approaches, can you provide more details?
2. While the paper provides a solid exploration of assignment and compactness loss functions, it would be insightful to see a deeper analysis of how these losses interact with other potential regularizers.
3. It would be beneficial to include a broader discussion on how prototypes evolve when user behavior changes over time or when new content is introduced. Realistic settings often involve rapidly shifting distributions, and observing how HVQ-VAE adapts (or could be adapted) would strengthen the paper’s practical insights.
4. The online experiment only includes results in Appendix, plz sharing more findings from an online A/B test would highlight how the proposed method translates to real-world gains, offering more confidence to practitioners.
5. This paper briefly touches on how prototype alignment can alleviate “representation confusion.” More extensive visualizations or case studies showing how specific user-video pairs move in representation space before and after alignment would further clarify this mechanism.
6. Adding a list of symbols can help others better understand the content.

**Questions For Authors:**

Please refer to the weakness.

**Relation To Broader Scientific Literature:**

This work applies vector-quantized variational autoencoders and distribution alignment via optimal transport to the recommendation context, extending beyond prior watch-time methods that rely on simplistic bucketization or direct regression. It proposes an effective scheme to tackle instance confusion via prototype calibration, particularly suited to scenarios featuring multi-modal label distributions.

**Theoretical Claims:**

Appendix A.1 provides the key mathematical derivation, proving that watch-time prediction error $\Delta$ is positively correlated with $\|f(x) - \mu_y\|$.  The proof views the network’s prediction as $\mathrm{ReLU}(W f(x) + b)$ and assumes there exists a center $\mu_y$ related to the true value. Under this assumption, closeness of f(x) to $\mu_y$ yields smaller prediction errors. The mathematical discussion, split according to the activation regions of ReLU, supports the approach of mitigating confusion by aligning representations to credible prototypes.

---

> ### Author Rebuttal · Authors · 2025-03-31
>
> We sincerely thank you for recognizing the significance of our work and for your generous positive-score evaluation. We are very grateful for your valuable suggestions and constructive questions, which have helped us improve this paper. Below, we provide detailed responses to your queries:
>
> > **Q1: modeling complexity(two-stage framework vs simple direct regression approaches)(refer to A.4)**
>
> Due to space constraints, please refer to the answer to Q3 in the rebuttal for Reviewer gDS8.
>
> > **Q2:  analysis of assign loss and compact loss**
>
> During Stage II, we adopt two losses to help with calibrating the sample space. In our method, we aims for the instance representations to cluster tightly around their corresponding prototypes. So the assign loss is designed to decrease the distance between samples and their corresponding prototypes. And since we hope for instances assigned to the same prototype to be closer together in the representation space, the compact loss is  designed to encourage samples under the same prototype to cluster more closely in the representation space
>
> By minimizing both losses, our method can not only help reduce instance representation confusion but also enhance the model’s ability to capture fine-grained features, ultimately improving prediction performance.
>
>
>
>
> > **Q3:  Model adjustment for changes in user behavior (refer to A.4)**
>
> We consider the following two aspects:
>
> (1) The stability of user behavior
>
> HVQ-VAE is completely independent, and its spatio-temporal complexity does not affect the training and inference time of ProWTP. When the distribution of watch ratios is sufficiently large, the resulting prototype distribution is stable. Furthermore, we observed that for a well-established video recommendation APP, the watch-ratio distribution remains largely unchanged and consistent across multiple months.
>
> As shown in Figure 10, we randomly sampled 200,000 users from our APP (a short-video platform) and extracted their historical behavior on the 1st day of each month from January to November 2024. The data were divided into D=15 buckets based on video duration. We then computed the Wasserstein Distance between the watch-ratio probability density distributions of each month and November, as well as the Kolmogorov-Smirnov test with $p<0.05$ between their cumulative empirical distributions. The results indicated no significant distribution shifts across multiple months.
>
> (2) Model adjustments driven by user behavior changes
>
> Even in extreme scenarios where user behavior undergoes notable adjustments, we only need to resample the watch-ratio distributions for each duration buckets, perform offline retraining, and update the weights of ProWTP. This process incurs minimal computational overhead.
>
> > **Q4: more findings from the online A/B test**
>
> |  Duration  |  0-7   |  8-12  | 13-24  | 25-42  | 43-60  | 61-84  | 85-120 | 121-160 | 161-320 | 321-600 |
> | :--------: | :----: | :----: | :----: | :----: | :----: | :----: | ------ | ------- | ------- | ------- |
> | watch time | 0.072% | 0.089% | 0.093% | 0.118% | 0.137% | 0.162% | 0.178% | 0.189%  | 0.213%  | 0.229%  |
>
> According to the results, we observed two key findings:
>
> 1. ProWTP demonstrated consistent performance improvements over D2Q across all duration buckets.
> 2. When examining different duration segments, we found that medium and long-form videos (>60s) exhibited significantly more pronounced watch time benefits compared to shorter videos. This pattern also aligns with the results observed in our offline dataset evaluations.
>
> > **Q5: alignment  visualization**
>
> We visualized the relationship between errors and instance representations in Appendix A.1 (Figures 4 and 5). By comparing these visualizations, we observed that:
>
> 1. Prediction error ∆ is positively correlated with the degree of confusion.
> 2. TR exhibits a significantly higher level of confusion, while ProWTP effectively mitigates this confusion by reducing the distance between instances and reliable prototypes.
> 3. Compared to TR, ProWTP shows significantly fewer points with large errors.
>
> This demonstrates that the root cause of reducing prediction errors lies in learning better instance representations.
>
> > **Q6: Symbol list**
>
> Thank you for your suggestion. Due to space constraints of rebuttal, we will add a list of symbols in the final paper.
>
> Thank you once again for your valuable suggestions and feedback. If you have any further questions, we would be happy to continue the discussion with you. We are looking forward to your reply.

---

> > ### Comment · Reviewer_3cw2 · 2025-04-03
> >
> > I appreciate the authors' thorough responses to all the questions. The additional experiments significantly strengthen the paper's validity: 1) real-world scenario tests demonstrate practical robustness, 2) hyperparameter sensitivity analyses clarify method stability, and 3) extended baseline comparisons provide broader contextualization. The discovery of multimodal phenomena rooted in real-world media data is particularly novel, offering fresh insights that could inspire new research directions in data-driven AI. These enhancements notably improve the paper's technical rigor and conceptual impact. Given the improved empirical validation and original findings, I raise my score to accept.

---

> > > ### Author Response · Authors · 2025-04-03
> > >
> > > Dear reviewer 3cw2,
> > >
> > > We sincerely thank you for your generous raise and recognition of our work and are honored to be able to address all of your concerns!

---

### Official Review · Reviewer_gDS8 · 2025-03-12

**Overall Recommendation:** 4

**Summary:**

This paper presents ProWTP, a novel two-stage approach for predicting user watch-time in video recommendation systems. The method improves prediction accuracy by aligning label distributions with instance representation distributions through prototype learning and optimal transport techniques. Specifically, it leverages a Hierarchical Vector Quantised Variational Autoencoder (HVQ-VAE) and Semi-relaxed Unbalanced Optimal Transport to address the inherent distribution characteristics of labels and mitigate instance representation confusion—challenges often overlooked by existing methods. Experimental results on three datasets demonstrate the effectiveness of ProWTP in enhancing watch-time prediction.

**Claims And Evidence:**

Yes, this paper has strong experimental and theoretical evidence to support the claim.

**Essential References Not Discussed:**

No essential references appear to have been omitted. The paper comprehensively cites and engages with relevant prior works.

**Experimental Designs Or Analyses:**

Yes, The experiments look solid. The ablation studies and comparative analyses effectively validate the contribution of each proposed component (HVQ-VAE, SUOT, assignment loss, compactness loss). More experiment results can be found in appendix.

**Methods And Evaluation Criteria:**

Yes, this paper used two public datasets and one private dataset with a reasonable training-testing-validation setup. The use of widely recognized metrics (MAE, RMSE, XAUC, XGAUC) ensures the results' validity and comparability with existing research.

**Other Comments Or Suggestions:**

* Minor typos are observed (e.g., "Sine" instead of "Since" in Section 5.1). Thorough proofreading is recommended.

* In Appendix, the authors have a typo: (see line 1074), A9 and 10 are the same thing

**Other Strengths And Weaknesses:**

S1: This paper presents a completely different perspective to optimize the WTP, proposing phenomena and methods that are very interesting and can be extended to more application scenarios.

S2: The hierarchical VQ-VAE-based credible prototype generation is innovative, providing clear advantages in capturing multimodal distribution patterns.

S3: Distribution alignment effectively mitigates the representation confusion issue, leading to improved predictive performance.

W1: Lack of detailed analysis regarding the sensitivity to hyperparameters, especially in different dataset scenarios.

W2: Minor language and grammatical errors should be revised for enhanced readability.

Overall: I think this paper is very interesting and the proposed phenomenon of multi-peaked distributions is valuable. And the scheme of conversion to prototype by distribution is very innovative.

**Questions For Authors:**

* Can authors discuss the computational efficiency of ProWTP, particularly in large-scale production environments?
* How sensitive is ProWTP to the hyperparameters (e.g., $\lambda$ in SUOT)? A brief analysis or guidance on hyperparameter tuning would enhance the practical applicability.
* Provide more online experiments environment will be better.
* How does ProWTP handle cases with highly imbalanced data distributions, such as scenarios with rare but critical user behavior patterns?

**Relation To Broader Scientific Literature:**

The ProWTP framework addresses a key limitation of existing watch-time prediction approaches, which typically transform watch-time labels for prediction and then reverse the transformation, neglecting both the natural distribution properties of labels and the instance representation confusion that can lead to inaccurate predictions.

Beyond improving the accuracy of video recommendations, this work bridges multiple subfields by offering insights into generalized regression problems, multimodal learning, and structured representation alignment in AI models. Its implications extend beyond recommendation systems, potentially influencing a broader range of machine learning applications that require distribution alignment and representation calibration.

**Theoretical Claims:**

Yes, I check the theoretical claims and can confirm the correctness.

---

> ### Author Rebuttal · Authors · 2025-03-31
>
> Dear Reviewer gDS8,
>
> We sincerely thank you for recognizing the significance of our work and for your generous positive-score evaluation. We are very grateful for your valuable suggestions and constructive questions, which have helped us improve this paper. Below, we provide detailed responses to your queries:
>
> > **Q1: Lack of detailed analysis regarding the sensitivity to hyperparameters**
>
> In our method, we have two hyperparameters: K (number of prototypes) and β (weight of the compact loss). In Section 5.2, we have already provided the experimental results and analysis for hyperparameter K. Due to space constraints, here we only supplement the hyperparameter β experiments on the KuaiRand-Pure dataset.
>
> | $\beta$ |  0.00  |  0.05  |    0.10    |  0.15  |  0.20  |
> | :-----: | :----: | :----: | :--------: | :----: | :----: |
> |  RMSE   | 41.12  | 40.66  | **40.45**  | 40.64  | 40.98  |
> |   MAE   | 24.92  | 24.70  | **24.43**  | 24.68  | 24.89  |
> |  XAUC   | 0.7221 | 0.7258 | **0.7290** | 0.7252 | 0.7223 |
> |  XGAUC  | 0.7004 | 0.7033 | **0.7048** | 0.7036 | 0.7001 |
>
> > **Q2: Minor language and grammatical errors**
>
> Thank you for pointing out the writing issues, we will correct it in the final paper.
>
> > **Q3: Complexity analysis and inference efficiency**
>
> Unlike simple direct regression approaches, our method incorporates an additional prototype generation process. But importantly, this prototype generation is completely independent, and its spatio-temporal complexity does not affect the training and inference time of ProWTP. When the distribution of watch ratios is sufficiently large, the resulting prototype distribution remains stable, which means that Stage I does not need to be performed frequently.
>
> In Stage II, during the inference phase, the OT module is removed. The final value is computed as a linear combination of similarities to each prototype, which is then input into the regressor. The time complexity of this process is only O(CK), where C and K are small constants, ensuring that the time overhead remains negligible.
>
>
>
>
> > **Q4: online experiments environments details**
>
> Thank you for your suggestion. Industrial recommendation systems typically employ a cascading architecture consisting of four stages: recall, pre-ranking, ranking, and re-ranking—a structure designed to efficiently recommend items to users from massive pools of video candidates. Since the recall stage operates without stringent real-time requirements, we strategically deployed ProWTP in this layer of an online short-video recommendation system, where it functions as one of multiple recall paths.
>
> To validate its effectiveness in real-world applications, we used D2Q as our baseline model for comparison. The experimental results convincingly demonstrate the significant performance improvements our method delivers in actual business scenarios.
>
> > **Q5: deal with imbalanced data distributions**
>
> Thanks for your question. In fact, video recommendation scenarios inherently face imbalanced data distributions in terms of video duration. Most video datasets, including WeChat and KuaiRand, exhibit left-skewed distributions in durations. Our experimental results clearly demonstrate that ProWTP outperforms competing models on the WTP task, showing particularly strong performance in watch time prediction under imbalanced data distribution conditions.
>
>
>
> Thank you once again for your valuable suggestions and feedback. If you have any further questions, we would be happy to continue the discussion with you. We are looking forward to your reply.

---

### Official Review · Reviewer_WSao · 2025-03-14

**Overall Recommendation:** 3

**Summary:**

The authors propose ProWTP, a two-stage method combining prototype learning and optimal transport for watch-time regression prediction and deep recommendation models. First, a hierarchical vector quantized variational autoencoder (HVQ-VAE) is used to convert the continuous label distribution into a high-dimensional discrete distribution, providing credible prototypes for calibration. Then, ProWTP views the alignment between prototypes and instance representations as a Semi-relaxed Unbalanced Optimal Transport (SUOT) problem, with the prototype constraints relaxed. Moreover, ProWTP introduces the assignment and compactness losses to encourage instances to cluster closely around their respective prototypes, enhancing the prototype-level distinguishability. Offline experiments on three industrial databases serve to demonstrate the proposed model performance.

## update after rebuttal
I acknowledge the authors for their rebuttal. The authors have addressed my main concerns, particularly about clarity in Figure 1, performance improvements, and the lack of qualitative analysis. The planned revisions to figures and the addition of prototype visualizations should significantly enhance the understanding of the proposed method. While the performance gains are modest, the authors have justified their practical significance, especially for medium-to-long videos. Given these improvements, I believe the paper could be accepted and have raised my score to Weak accept.

**Claims And Evidence:**

The authors start motivating their research by highlighting the need for video recommendation systems for personalized content, where the user watch-time metric constitutes a key metric for measuring user engagement. They state that existing Watch-Time Prediction (WTP) models have difficulties in achieving a high predictive accuracy across different recommendation systems because they do not consider the multimodal distribution of labels, reflected in Figure 1(a).

Then, the authors also highlight instance representation confusion. Although this issue is illustrated in Figure 1(b), neither a description of the features represented in the 2D scatter plot nor the concept itself (i.e., the authors only refer to “various patterns”) is provided, which makes it difficult to understand this claim.

When reaching the Related Work section, i.e., Section 2, the authors properly identify additional drawbacks of existing WTP approaches (e.g., difficult to model ordinal relationships and quantiles, no uncertainty quantification). They also review optimal transport and deep clustering with variational autoencoders (VAEs), identifying those works that constitute the basis of the proposed ProWTP, i.e., Semi-relaxed Unbalanced Optimal Transport (SUOT, see subsequent Section 3. Background) and Vector Quantized Variational Autoencoder (VQ-VAE), respectively.

Overall, the claims made in the paper seem to make sense, but there is a lack of details on the evidence provided in Figures 1(a) and 1(b). Conversely, it should be noted that the Related Work and Background sections are complete and comprehensive, allowing for researchers without experience in WTP, optimal transport, and deep clustering to understand the core of the proposed methodology.

**Essential References Not Discussed:**

Overall, the related works mentioned in the paper seem relevant and enough to understand the context for the main contributions of the paper. Both Related Work and Background sections are complete and comprehensive, identifying advantages and limitations of existing methods.

**Experimental Designs Or Analyses:**

Reported results include first a comparison with several baselines representing popular WTP in the literature. However, according to Table 1, it should be noted that ProWTP slightly outperforms most of these approaches. A qualitative/error analysis could have been helpful to get a better idea about which situations the proposed method is more advantageous than existing ones, thanks to prototypes and SUOT. The visualization/analysis of the prototypes learned could have also significantly strengthened the discussion of results.

Second, an ablation study to measure the impact of the different ProWTP modules is shown in Table 2, supporting the suitability of the design decisions made. Third, different prototype generation methods and distribution alignment methods are tested, reporting results in Tables 3 and 4 to validate the usefulness of the proposed HVQ-VAE and SUOT. It should be noted that the improvements shown are again very small overall, even when compared to classical and simpler K-means/Random methods for clustering, or not using alignment for distribution. The authors should have completed their analysis by focusing on those samples where ProWTP has a clear advantage. Finally, the impact of the number of prototypes learned is shown in Figure 3, again with little differences appreciated between configurations. Although the proposed methodology for ProWTP is convincing, results do not seem to support its significance.

**Methods And Evaluation Criteria:**

The proposed method for watch-time prediction is properly designed to overcome the limitations of existing approaches identified at the beginning of the paper. First, the authors make use of prototypes for improving prediction given the multimodal distribution followed by watch-ratio. These prototypes are generated directly from the distribution of the prediction target via the proposed hierarchical VQ-VAE module. Second, to mitigate instance representation confusion, the generated prototypes are used to calibrate instance representations via alignment using SUOT. Finally, training objectives/loss functions are introduced, with additional details provided in the Appendix, important for the sake of reproducibility.

About the evaluation criteria, the authors make use of three different databases (Wechat, Kuairand, and Short-video), suitable for the application to be addressed. The performance of the proposed ProWTP is evaluated using well-known metrics such as MAE, RMSE, and AUC extensions for continuous values. Several baselines representing popular WTP methods are considered for comparison purposes.

In summary, the proposed methods and evaluation criteria make sense for the WTP task.

**Other Comments Or Suggestions:**

- The authors should have referred to the additional details provided in the Appendix throughout the paper, to ensure readers have a clearer understanding of those concepts which are briefly described in the paper, e.g., instance representation confusion.
- Figure 1(a): Which database(s) have been used to estimate and show the watch-ratio distribution? A brief description of the nature/contents of the videos considered should be provided at least for better understanding.
- Figure 1(b): How representation confusion is represented, i.e., which database(s)/features were used for the scatter plot?
- Figure 1(c) and Figure 2: The quality of the diagrams shown to illustrate the proposed ProWTP could be notably improved, for better comprehension. I think they should be less schematic and more specific on the details of the different stages (watch-ratio distribution, W, meaning of codebook vectors and matrices, etc.). Try to align them with the ProWTP description provided in Section 4 as much as possible.
- Figure 1 caption should be descriptive, allowing for the understanding of the figure without the need for reviewing the corresponding paragraphs in the paper.
- Appendix A.1.1: The proposition for instance representation confusion seems correct and easy to understand, but the authors should have provided details on the experiments conducted to verify that only a small percentage of the total training data lies in the non-activation region of ReLU.

**Other Strengths And Weaknesses:**

The proposed ProWTP method effectively integrates prototype learning with optimal transport in a novel way to enhance watch-time prediction (WTP), addressing key challenges related to prediction accuracy and representation confusion. The paper is well-organized, including a thorough literature review and a clear methodological presentation, which makes it accessible to researchers beyond WTP experts. The evaluation is robust, incorporating multiple datasets and diverse baselines to ensure the reliability of the reported results. Additionally, the supplementary material is comprehensive, covering aspects such as computational complexity and additional experimental validations.

However, the reported performance improvements over baselines are relatively small, raising concerns about the practical impact of ProWTP. The experimental section could benefit from qualitative analyses, such as visualizations of learned prototypes and error analyses, to better illustrate when and why ProWTP is most effective. Finally, the clarity of the methodology figures (Figures 1 and 2) could be improved to better align with the textual descriptions, ensuring that the proposed approach is easily understandable.

Based on the strengths and weaknesses of the paper, my current overall recommendation is “Weak reject”.

**Questions For Authors:**

Q1. The reported improvements over baselines are relatively small. Could you provide additional justification for the practical significance of ProWTP? Are there specific scenarios where these gains are particularly meaningful?

Q2. Have you considered including qualitative analyses, such as visualizations of learned prototypes or error breakdowns, to better illustrate when and why ProWTP outperforms other methods? Could such analyses identify specific cases where ProWTP is most effective?

**Relation To Broader Scientific Literature:**

The key contributions of the paper integrate concepts from several research areas, including VAEs, deep clustering, and optimal transport, for watch-time prediction, to enhance this task in recommendation systems. The use of hierarchical VQ-VAEs for discrete representation of continuous watch-time distributions aligns with recent advances in prototype learning and quantized representation learning. Moreover, the use of SUOT for aligning instance representations with prototypes extends prior research on transport-based methods for structured prediction tasks.

**Theoretical Claims:**

A description and a mathematical explanation of instance representation confusion are provided in Appendix A.1. The authors should have referred to this additional information in the main paper, to ensure readers understand this less-known concept. The corresponding proposition seems correct and easy to understand, but the authors do not provide details on the experiments conducted to verify that only a small percentage of the total training data lies in the non-activation region of ReLU.

---

> ### Author Rebuttal · Authors · 2025-03-31
>
> Dear reviewer WSao,
>
> We greatly appreciate your valuable questions and suggestions, which have helped us improve this paper significantly. We summarize your concerns below and provide detailed responses.
>
> > **Q1: Details of Figures 1 and 2, and subsequent modifications**
>    - Figure 1(a) shows the watch-ratio distributions of three different video duration buckets (short/mid/long) in the Wechat dataset.
>    - Figure 1(b) provides a dimensionality-reduced visualization of the instance representations $f(x)$ generated by the TR model on the Wechat dataset, using samples from four different video duration buckets to illustrate “instance representation confusion.” Since this figure does not fully convey the core issue of instance representation confusion, we plan to replace it with Figure 4(4) (from the Appendix) in the revised paper to present a clearer illustration in the Introduction.
>    - Figure 2 depicts the two-stage training process of ProWTP:
>      (1) In the first stage, the watch-ratio distribution $W$ is used as input and reconstructed via the HVQ-VAE. We then extract the parameter $P$ (the codebook) obtained in this process to serve as the prototype parameter for the second stage.
>      (2) In the second stage, sample features $X$ are used as input, and the output is the predicted $y$.
>
>    We will enhance Figure 2 with more symbols and annotations in our next revision for better clarity.
>
> ---
>
> > **Q2: Experimental improvements and their significance in real-world scenarios**
>
>    - Table 1 reports the average performance over five runs for each model. Generally, for ranking metrics like AUC, an improvement of 0.001 is considered significant. The improvement of 0.01 in RMSE and MAE metrics is significant, which is consistent with previous studies [1].
>    - Our online experiments indicate that different video duration buckets exhibit varying levels of watch-time gain, with **medium-to-long videos** seeing the greatest benefits. This outcome suggests that ProWTP can deliver higher gains in practice by recalibrating samples that are more prone to large prediction errors (i.e., medium and long videos).
>
> |  Duration(s)  |  0-7   |  8-12  | 13-24  | 25-42  | 43-60  | 61-84  | 85-120 | 121-160 | 161-320 | 321-600 |
> | :--------: | :----: | :----: | :----: | :----: | :----: | :----: | ------ | ------- | ------- | ------- |
> | watch time | 0.072% | 0.089% | 0.093% | 0.118% | 0.137% | 0.162% | 0.178% | 0.189%  | 0.213%  | 0.229%  |
>
> [1] CREAD: A Classification-Restoration Framework with Error Adaptive Discretization for Watch Time Prediction in Video Recommender Systems.
>
> ---
>
> > **Q3: Lack of prototype visualization and error analysis**
>
>    - We observed that the prototypes learned by ProWTP form distinct “striped” clusters in high-dimensional space, and we will add a prototype visualization in the revised version.
>    - As mentioned in the previous section, ProWTP is particularly effective for medium-to-long videos, which inherently have larger prediction errors. By using prototypes as anchors and bringing samples with large errors closer, the overall performance is improved. Below is RMSE comparisons between TR and ProWTP for different video-duration buckets in the Wechat dataset to illustrate this improvement. According to the proof of A.1, the larger the prediction error, the further its f (x) is from the ideal center, then the larger ASSIGNMENT LOSS will be, and ProWTP will focus on optimizing such samples.
>
> | Bucket | TR    | ProWTP |
> |--------|-------|--------|
> | 0      | 8.62  | 8.57   |
> | 1      | 13.74 | 13.13  |
> | 2      | 21.58 | 20.15  |
> | 3      | 33.42 | 31.95  |
> | 4       | 52.62 | 48.97  |
> ---
> > **Q4: Explanation in A.1.1 regarding why only a small portion of training data falls into the non-activation region of ReLU**
>
>    - We tested all three datasets; for each sample $(x, y)$, $y$ is the ground truth, and $ReLU(model(x))$ is the prediction. We found that the number of samples with \(model(x) < 0\) (i.e., the activation output being zero) accounted for only about 1–2% of the total data in each dataset.
>    - This suggests that very few samples lie in the non-activation region of ReLU, supporting our assumption about instance confusion in Appendix A.1. We will include a scatter plot in subsequent revisions to further verify this claim.
>
> ---
>
> > **Q5: Revisions to the main text and Appendix linkage to improve readability**
>
> We appreciate your suggestions and plan to reference Proposition A.1 in Section 4.3. At the same time, we will revise Figure 1(b) to more clearly explain the concept of instance representation confusion.   By doing so, we hope to guide readers to the relevant sections of the Appendix and ensure they can easily find the in-depth explanations presented there.
>
> ---
>
> We sincerely thank you again for your feedback and hope that our responses can change your mind. We look forward to your reply and further discussion.

---

### Decision · Program_Chairs · 2025-05-01

**Decision:**

Accept (poster)

**Comment:**

This paper presents a novel, well-motivated method for improving video watch-time prediction via prototype-based calibration and structured alignment. The approach is theoretically well-founded, empirically validated on large-scale datasets, and deployed in a real-world industrial system. While quantitative improvements are modest, the novelty and deployment practicality offer considering contributions to application-driven ML.

### Strengths
- Clear Motivation and Mathematical Justification: The paper addresses “instance representation confusion”—a unique perspective on model error origins—and provides a theoretical proposition linking prediction error to prototype alignment. Reviewers appreciated the clarity and rigor of this formulation.
- Method Novelty: The paper presents a novel two-stage framework (ProWTP) that integrates prototype learning with optimal transport (OT) for calibrated watch-time prediction. By leveraging hierarchical vector-quantized VAEs (HVQ-VAE), it transforms continuous labels into discrete prototypes, which are aligned with instance representations using semi-relaxed unbalanced OT. This structure is both theoretically grounded and practically effective.
- Extensive Experiments: The ProWTP is evaluated across three real-world industrial datasets, with thorough ablation studies, hyperparameter sensitivity analyses, and online A/B tests. These demonstrate both empirical efficacy and deployment readiness. The approach shows particular benefit on medium-to-long videos, which typically suffer from higher prediction variance.

Considering minor presentation issues and one problematic reviewer, the majority of reviewers lean to acceptance, with at least two increasing their scores post-rebuttal.